# The molecular basis of sugar detection by an insect taste receptor

João Victor Gomes[1], Shivinder Singh-Bhagania[1], Matthew Cenci[1], Carlos Chacon Cordon[1], Manjodh Singh[1] & Joel A. Butterwick[1✉]

Animals crave sugars because of their energy potential and the pleasurable sensation of tasting sweetness. Yet all sugars are not metabolically equivalent, requiring mechanisms to detect and differentiate between chemically similar sweet substances. Insects use a family of ionotropic gustatory receptors to discriminate sugars[1], each of which is selectively activated by specific sweet molecules[2–6]. Here, to gain insight into the molecular basis of sugar selectivity, we determined structures of Gr9, a gustatory receptor from the silkworm *Bombyx mori* (BmGr9), in the absence and presence of its sole activating ligand, D-fructose. These structures, along with structure-guided mutagenesis and functional assays, illustrate how D-fructose is enveloped by a ligand-binding pocket that precisely matches the overall shape and pattern of chemical groups in D-fructose. However, our computational docking and experimental binding assays revealed that other sugars also bind BmGr9, yet they are unable to activate the receptor. We determined the structure of BmGr9 in complex with one such non-activating sugar, L-sorbose. Although both sugars bind a similar position, only D-fructose is capable of engaging a bridge of two conserved aromatic residues that connects the pocket to the pore helix, inducing a conformational change that allows the ion-conducting pore to open. Thus, chemical specificity does not depend solely on the selectivity of the ligand-binding pocket, but it is an emergent property arising from a combination of receptor–ligand interactions and allosteric coupling. Our results support a model whereby coarse receptor tuning is derived from the size and chemical characteristics of the pocket, whereas fine-tuning of receptor activation is achieved through the selective engagement of an allosteric pathway that regulates ion conduction.

Sugars are a primary source of energy and influence nutrient sensing, metabolic responses, reward mechanisms and taste perception[7,8]. Most animals can taste sugars through sweet receptors expressed in dedicated taste organs[9]. Mammals taste all sweet compounds using a single heterodimeric taste receptor, composed of two G-protein-coupled receptors (T1R2 and T1R3)[10]. Insects, instead, rely on a distinct family of sweet gustatory receptors[1], which are tetrameric ligand-gated cation channels[2]. A distinctive feature of these gustatory receptors is that each receptor specializes in detecting only a subset of sugar molecules[2–6], despite the high degree of chemical similarity among saccharides. An extreme example of sugar specialization occurs in a conserved subfamily of gustatory receptors founded by *Drosophila melanogaster* Gr43a (DmGr43a), whose members are selectively activated only by the monosaccharide D-fructose[2,3], providing a unique opportunity to investigate how specificity for a single sugar is achieved.

Here we investigate the structural basis of sugar discrimination by Gr9 (BmGr9), the silk moth *B. mori* orthologue of DmGr43a. We determined structures of BmGr9 in two gating states: closed, in the absence of sugar; and opened, in the presence of D-fructose. BmGr9

harbours a sugar-binding pocket in the transmembrane region of each subunit that tightly envelopes D-fructose and precisely coordinates its hydroxyl groups through a set of conserved polar amino acids. Despite these seemingly specific interactions, computational docking and experimental binding studies suggest that other sugar molecules also fit into the BmGr9 pocket, but they are unable to activate the receptor. Thus, the geometric arrangement of chemical groups inside the ligand-binding pocket does not seem to be sufficient to explain the selective activation of this receptor by only D-fructose. We therefore determined the structure of BmGr9 in a third gating state bound to a non-activating sugar, L-sorbose, but with a closed pore. The bound sugar is similarly positioned in the ligand-binding pocket to D-fructose, but does not induce a conformational change in a bridge of two conserved aromatic residues that is required to open the channel gate. Activation efficacy, therefore, depends on residues that extend beyond the receptor–ligand interactions that occur in the binding pocket. Our findings show how narrow chemical tuning can be achieved in taste receptors and suggest how the tuning of chemoreceptors could be adjusted to recognize different regions of chemical space.

[1]Department of Pharmacology, Yale University School of Medicine, New Haven, CT, USA. ✉e-mail: joel.butterwick@yale.edu

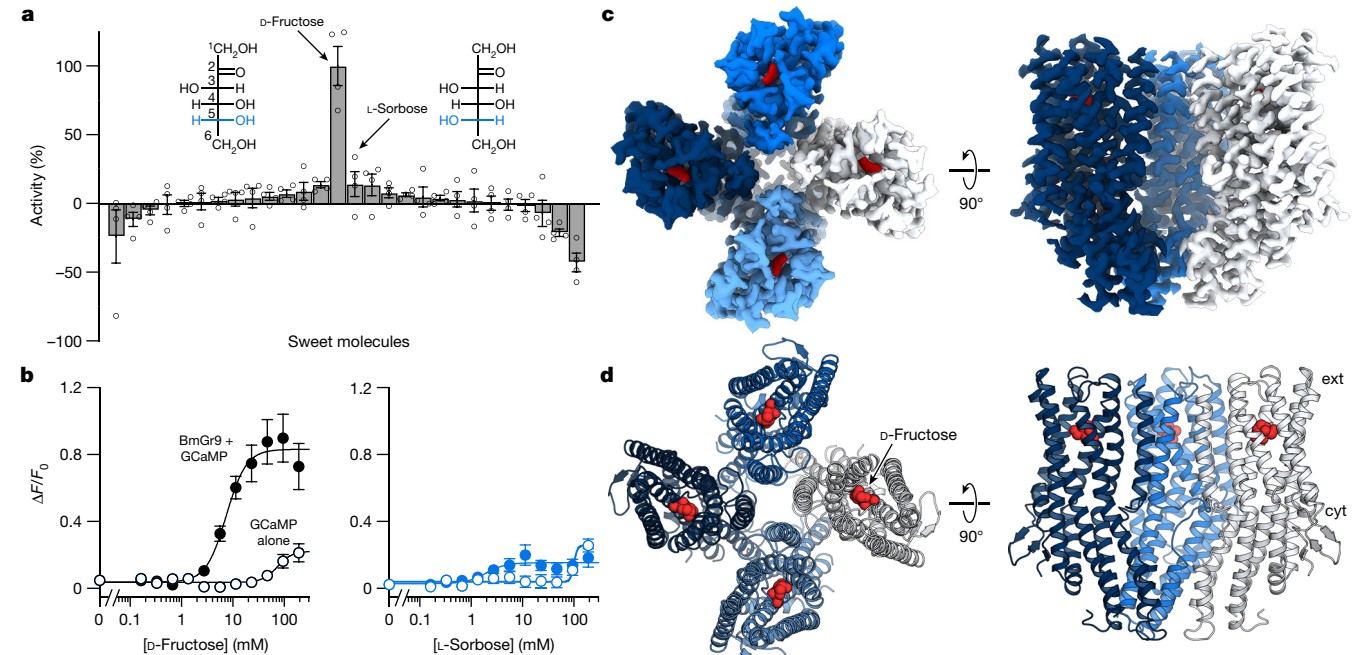

**Fig. 1 | Structure of an insect taste receptor. a**, Activation of BmGr9 by a panel of sweet compounds, measured as the change in fluorescence relative to that for D-fructose. Bars are mean ± s.e.m. with independent samples ($n$ = 4) shown as open circles. Insets show Fischer projections of the epimers D-fructose and L-sorbose, with carbons numbered and difference highlighted in blue. **b**, Dose–response of raw fluorescence changes of HEK293 cells transfected with BmGr9 and GCaMP (filled circles) or GCaMP alone (open circles) when titrated with D-fructose (left) or L-sorbose (right). D-Fructose data are best fitted by an $EC_{50}$ of 8.2 (5.9–11.5) mM with a Hill coefficient of 2.3 (1.3–4.6) ($n$ = 8 independent samples; points are mean ± s.e.m.; fitted 95% confidence intervals are given in parentheses). For L-sorbose ($n$ = 5 independent samples), the maximum activity is less than that for control wells with GCaMP alone. **c,d**, Cryogenic electron microscopy density map (**c**) and ribbon model (**d**) of BmGr9 bound to D-fructose (red) shown from the top (left) and side (right). Approximate boundaries for the extracellular (ext) and cytoplasmic (cyt) sides are indicated. In **c,d**, the front subunit has been removed from the side views to expose the pore.

## Structure of BmGr9 bound to D-fructose

We confirmed the narrow chemical tuning of BmGr9 by transiently co-expressing the receptor with a fluorescent calcium reporter, GCaMP6s[11], in HEK293 cells and recorded fluorescence changes following addition of sweet compounds. Of a panel of 27 naturally occurring sugars and artificial sweeteners, we found that only D-fructose elicited a strong response (Fig. 1a and Extended Data Fig. 1), which is consistent with previous results[2]. Fitting the Hill equation to dose–response data yielded a half-maximal activation concentration ($EC_{50}$) of 9 mM for D-fructose (Fig. 1b). Such low affinities are common for sugar-sensing gustatory receptors[2,5] and probably reflect the high concentrations of sugars typically found in floral nectars[12] and insect haemolymph[3]. Notably, even sugars that are highly structurally similar to D-fructose, such as L-sorbose (an epimer of fructose differing only by the relative orientation of a single hydroxyl group), did not activate BmGr9 significantly (Fig. 1a,b).

To investigate the structural basis for the remarkable sugar specificity of BmGr9, we purified the receptor in the presence of a saturating amount of D-fructose (Extended Data Fig. 2a,b) and determined the structure of the complex using single-particle cryogenic electron microscopy. Three-dimensional reconstruction with four-fold averaging yielded a density map with 3.0 Å overall resolution (Fig. 1c, Extended Data Fig. 2c–f and Extended Data Table 1), which allowed us to build a model for most of the protein (Fig. 1d). The structure of BmGr9 closely resembles that of insect olfactory receptors[13,14], demonstrating how the same overall architecture underlies detection of tastants and odorants in insects, unlike the case for mammalian receptors, which recognize these compounds through distinct families using different binding domains[15].

## Sugar-binding pocket

D-Fructose is predicted to bind within an extracellular-facing pocket formed by the S1–S6 transmembrane helices of each BmGr9 subunit[16]. We observed additional density in the putative sugar-binding pocket, not attributable to protein, that is the approximate size and shape of a monosaccharide. In aqueous solution, D-fructose rapidly interconverts between five-membered (α- and β-furanose) and six-membered (β-pyranose) ring configurations[17,18]. To help determine which form of D-fructose is bound in our structure, we computationally docked fructose conformers into the pocket using AutoDock Vina[19,20]. We found that the β-furanose and β-pyranose forms docked with the lowest energy scores ($-5.7$ kcal mol$^{-1}$), whereas the calculated energetics of α-furanose was less favourable ($-5.0$ kcal mol$^{-1}$; Extended Data Fig. 3a–c). We, therefore, refined a model of BmGr9, bound to both β-D-fructopyranose and β-D-fructofuranose, with atom occupancies set to their anomeric distribution in solution. The refinement yielded nearly identical ligand conformations (Extended Data Fig. 3d–f). In what follows, we focus on β-D-fructopyranose, as it is the main conformer found in solution (about 75%)[17,18], but both forms seem capable of binding to and activating BmGr9, and the observed density is probably a superposition of furanose and pyranose ring forms (Extended Data Fig. 3f).

The sugar-binding pocket in BmGr9 extends from the extracellular surface to almost halfway through the membrane. D-Fructose sits at the base of this pocket, approximately 15 Å from the extracellular surface, making direct contact with residues in helices S2–S6 (Fig. 2a,b and Extended Data Fig. 3h,i). D-Fructose is oriented such that its hydrophobic and hydrophilic surfaces make very different interactions with residues within the binding pocket. Notably, one of the hydrophobic faces of β-D-fructopyranose (and β-D-fructofuranose), formed by hydrogens

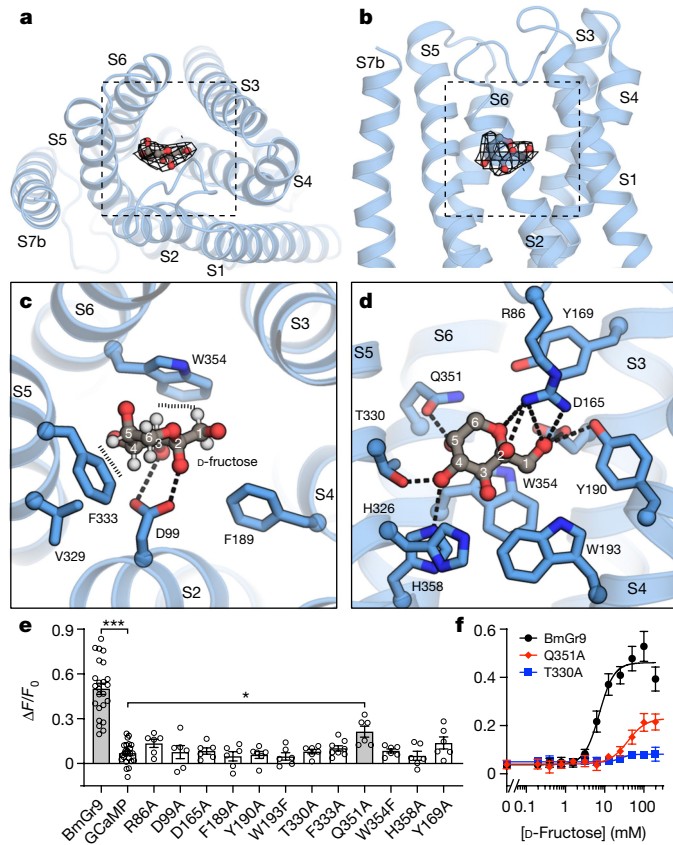

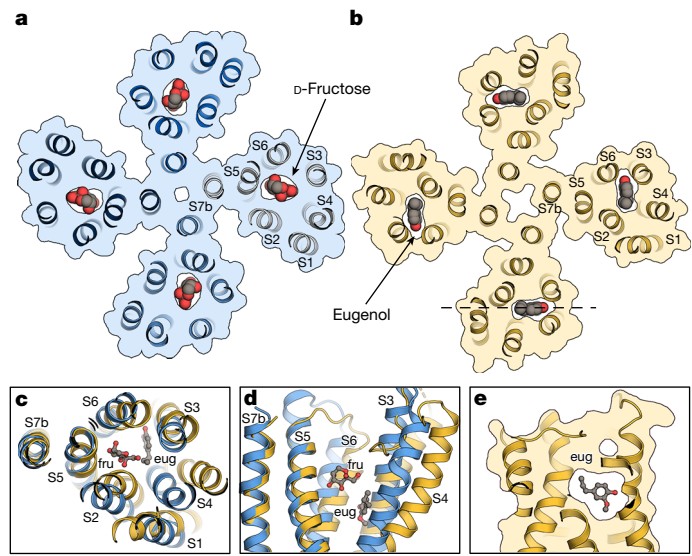

**Fig. 2 | Sugar-binding pocket in BmGr9. a,b,** Position of β-D-fructopyranose (grey, carbons; red, oxygens) within a subunit of BmGr9 (blue) viewed from the top (**a**) and side (**b**). **c,d,** Close-up views showing interactions between BmGr9 and D-fructose (with carbons numbered). Polar interactions are drawn as dashed lines. In **c**, hydrogens on D-fructose (white) are shown to highlight hydrophobic interactions with Trp354 and Phe333 (indicated by vertical dashes). **e,** Effect of substitutions within the sugar-binding pocket on the activity of BmGr9. Bars are mean ± s.e.m. with independent samples shown as open circles. Only wild-type BmGr9 and Q351A have significantly different activity from that of GCaMP alone (indicated by grey bars). Statistical significance was determined using one-way analysis of variance (ANOVA) followed by Dunnett's multiple comparison tests (***$P < 0.0001$, *$P = 0.035$). **f,** Dose response of select mutants from **e**. Data points are mean ± s.e.m. from $n = 6$ (Q351A and T330A) or 8 (wild-type BmGr9) independent samples measured from the same plates.

on C1 and C3, borders Trp354 in S6 at the side of the pocket (Fig. 2c). In protein regions that interact with sugar molecules, tryptophan and other aromatic residues are common owing to the favourable formation of CH–π interactions[21]. The distinct hydrophobic regions of each sugar dictate the preferred orientation of the molecule when bound to its cognate receptor[21]. Thus, the Trp–fructose interaction probably positions the sugar within the binding pocket.

The hydroxyl groups in D-fructose are all coordinated by polar groups (Fig. 2d), from acidic (Asp99 and Asp165), basic (Arg86 and His358) and uncharged (Tyr190, Trp193, Thr330 and Gln351) residues. Individual hydroxyls often interact with multiple amino acids, creating a network of bridges between transmembrane helices. For example, hydroxyl groups on C1 (bridging S3 and S4), C3 (S2 and S4) and C4 (S5 and S6) all link neighbouring helices, probably imparting considerable stability to the pocket when sugar is present.

Amino acids lining the pocket are highly conserved among fructose-selective receptors, but not in other sweet-sensing gustatory receptors (Extended Data Fig. 4), providing a rationale for the differing sugar sensitivities among gustatory receptors. Substitution

of the aromatic and polar residues within the pocket to Ala (or Trp to Phe) eliminated activation by D-fructose, with the exception of Gln351 to Ala, which retained only marginal activity (Fig. 2e,f). More conservative substitutions were also generally not tolerated. For instance, mutating either Asp99 or Asp165 to Asn, Gln or Glu resulted in non-functional channels, but Arg86 to Lys retained activity (Extended Data Fig. 5 and Extended Data Table 2). The strict requirement for specific amino acids suggests that their distribution within the binding pocket of BmGr9 forms a precise geometric arrangement to coordinate D-fructose.

**Fig. 3 | Ligand-binding locus is conserved in insect chemoreceptors. a,b,** Slices through the pockets of BmGr9 bound to D-fructose (**a**) and MhOr5 bound to eugenol (**b**; Protein Data Bank: 7LID) highlighting the positions of their respective ligands. The dashed line in **b** denotes the location of the slice presented in **e**. **c,d,** Superposition of a single subunit of BmGr9 (blue) and MhOr5 (gold) showing the relative positions of D-fructose (fru) and eugenol (eug), viewed from the top (**c**) and side (**d**). **e,** A vertical slice through the pocket of MhOr5, in a similar orientation to the structures in Fig. 4b,c, showing eugenol encapsulated by MhOr5 with no direct means of egress.

## Conserved ligand-binding locus

Comparing the structure of fructose-bound BmGr9 to an eugenol-bound olfactory receptor from the jumping bristletail *Machilis hrabei* (MhOr5)[14] shows that sugar and odorant occupy similar locations (Fig. 3a,b), suggesting that the ligand-binding locus is conserved across the insect chemoreceptor superfamily. However, BmGr9 and MhOr5 contrast in several critical aspects. Notably, sugar and odorants occupy slightly different regions of the pocket. D-Fructose sits close to the pore at the inner edge of the pocket, partially exposed to the extracellular solution and interacting closely with many residues along S5 and S6 (Fig. 3c,d). Eugenol, instead, sits in an occluded cavity adjacent to S3 and S4 at the outer edge, approximately 6 Å distal and 6 Å deeper than fructose, too far to directly interact with S5. In the presence or absence of an odorant, the ligand-binding pocket of MhOr5 is enclosed by protein[14] (Fig. 3e). Hydrophobic odorants have been suggested to enter the binding pocket from the membrane through a tunnel formed transiently between S3 and S6, which would allow odorants to access the pocket near the outer edge[22]. As water-soluble sugars seem able to enter the pocket directly from the extracellular space, the positional differences of the ligands observed here may represent a distinctive feature of gustatory receptors versus olfactory receptors.

The structural and chemical characteristics of the ligand-binding pockets are also different between BmGr9 and MhOr5. BmGr9 binds its single ligand with a hydrophilic pocket built to recognize the specific

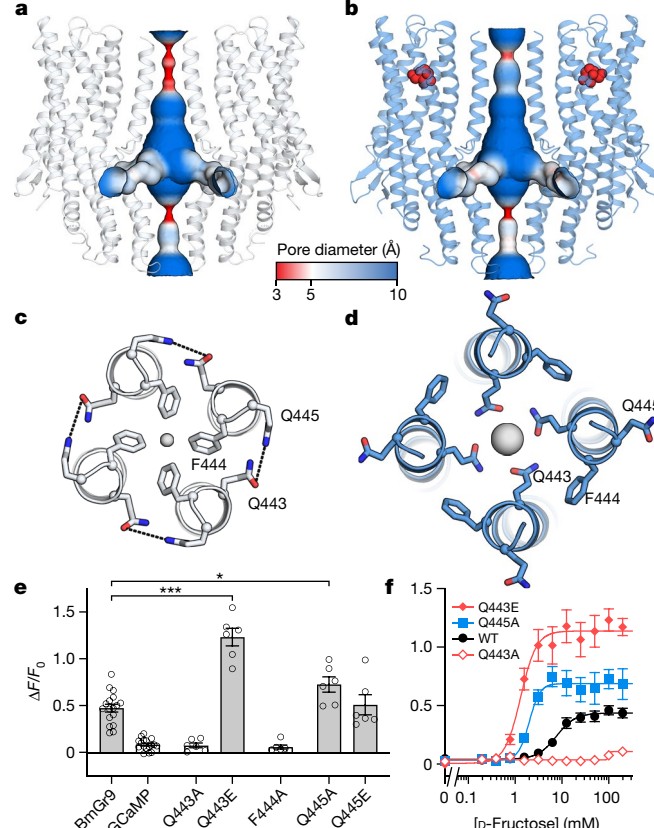

**Fig. 4 | Gating of BmGr9. a,b**, The ion permeation pathway of BmGr9, coloured according to pore diameter, in the absence (**a**) or presence (**b**) of D-fructose (red spheres). **c,d**, Close-up views of the pore helices in the absence (**c**) or presence (**d**) of D-fructose shown from the top, highlighting key residues in the wetting transition of BmGr9. Hydrogen bonds between Gln443 and Gln445 of adjacent subunits in the closed state are indicated by black dashed lines. Central grey spheres illustrate the narrowest pore diameter, near Phe444 (1.2 Å; **c**) or Gln445 (3.1 Å; **d**). **e**, Fluorescence changes of BmGr9 and mutants when stimulated with 100 mM D-fructose. Q443E and Q445A substitutions yield channels that are more active than those of the wild-type BmGr9. Bars are mean ± s.e.m. with independent samples shown as open circles; grey bars indicate statistically significant activity compared to that of GCaMP alone. Statistical significance was determined using one-way ANOVA followed by Dunnett's multiple comparison tests (***$P < 0.0001$, *$P = 0.012$). **f**, Dose–response curves of select mutants compared to wild-type BmGr9. Data points are mean ± s.e.m. from $n = 6$ (Q443A, Q443E and Q445A) or 12 (wild type) independent samples measured from the same plates.

molecular features of sugars, whereas MhOr5 engages diverse odorants with a promiscuous hydrophobic pocket that accommodates differently shaped molecules[14], explaining the vastly different tuning profiles of these two chemoreceptors.

## Gating and cooperativity among subunits

To gain insight into the mechanism of receptor activation, we also determined the structure of BmGr9 in the absence of ligand. We resolved a map of similar quality as the fructose-bound BmGr9 (Extended Data Fig. 2g–j) but with the extracellular hydrophobic gate closed, consistent with the absence of any density in the ligand-binding pocket (Fig. 4a,b). In the unbound structure, Phe444 side chains from each subunit face the centre of the ion-conducting pore, erecting a hydrophobic barrier that constricts the pore diameter and prevents ion passage (Fig. 4c), similar to the gate observed in olfactory receptors[13,14]. When D-fructose binds, these hydrophobic groups swing anticlockwise away from the

pore and are replaced by the polar side chains of Gln443. These rearrangements, accompanied by outward movements of the extracellular ends of the S7b pore helices, result in a widened pore with hydrophilic character (Fig. 4d). Gln443 and Phe444 are part of the only signature sequence found in the insect chemoreceptor superfamily (TYhhhhhQF, in which h is any hydrophobic amino acid)[23,24], and mutating either residue to Ala inactivated the receptor (Fig. 4e,f). However, substitution of Gln443 to Glu resulted in enhanced activity and increased apparent affinity for D-fructose, suggesting that a negatively charged residue at this position favours the activated state of the channel. A similar hydrophobic-to-hydrophilic 'wetting' transition was also observed in MhOr5[14] (Extended Data Fig. 6a,b), suggesting that it is probably a common feature of insect chemoreceptor gating.

In the closed state, Gln443 hydrogen bonds with Gln445 from a neighbouring subunit (Fig. 4c). Eliminating this interaction by mutating Gln445 to Ala created a channel with increased activity and higher apparent affinity compared to that of the wild-type receptor (Fig. 4e,f). Gln445 alterations that retain hydrogen-bonding potential (to Glu or Asn), however, yielded activity similar to that of wild-type channels (Extended Data Fig. 7 and Extended Data Table 2). Thus, without Gln445 to stabilize the position of Gln443 outside the pore, Gln443 can swing into the pore more easily. This interaction illustrates a possible mechanism for gating cooperativity within the tetramer: ligand binding to one subunit will induce movement of the pore helix, S7b, breaking the Gln443–Gln445 interaction and thus freeing Gln443 of the neighbouring subunit to face the pore. We reassessed the structures of Orco and MhOr5 and identified similar intersubunit interactions, between Tyr466 and Gln472 in Orco[13] and Asn469 and Gln467 in MhOr5[14] (Extended Data Fig. 6b,c), suggesting that S7b–S7b intersubunit hydrogen-bonded connections are conserved among this chemoreceptor superfamily.

## An aromatic bridge links pocket and pore

Comparing the bound and unbound structures of BmGr9 reveals a series of helix movements and side-chain reorientations that occur following D-fructose binding. Helices S1, S3, S4 and S6 remain largely fixed, whereas S2 and S5 move to constrict the pocket around D-fructose (Fig. 5a). These conformational changes seem to be driven by specific interactions with D-fructose: Asp99 in S2 moves about 3 Å to hydrogen bond with the hydroxyl group on C2, and Phe333 in S5 moves about 2 Å to interact with a second hydrophobic face in β-D-fructopyranose, formed by hydrogens on C4, C5 and C6 (C4 and C6 in β-D-fructofuranose). In the absence of a ligand, the vestibule connecting the pocket to the extracellular surface is approximately 8 Å wide, large enough for sugar molecules to access the bottom of the pocket freely (Fig. 5b). Following binding, the tunnel shrinks to 3 Å wide, reducing the pocket volume by about half and tightly enveloping the sugar (Fig. 5c).

Between the pocket and the pore sit Tyr332 and Phe333, which form an 'aromatic bridge' on S5 that directly connects D-fructose and residues on the S7b pore helix (Fig. 5a). The concerted movement of these residues when D-fructose binds may serve as a switch to open the pore: as Phe333 shifts towards D-fructose, the adjacent aromatic residue Tyr332 is pulled away from S7b, creating space for the pore helix to move. Tyr332 and Gln445 share a hydrogen bond in both the unbound and bound states, maintaining the direct link between the positions of S5 and S7b. Substitution of Tyr332 to Phe slightly increases the apparent affinity for D-fructose, whereas substitution to Ala or Leu decreases or eliminates activity, respectively (Fig. 5d,e). These varied responses with different amino acids at position 332 suggest that the S5–S7b connection can be fine-tuned to modulate pore opening, regardless of specific receptor–ligand interactions within the pocket. By contrast, substitution of Phe333 to Ala, Leu or Tyr significantly reduced or eliminated activity. This intolerance to modification may reflect the additional need for Phe333 to directly engage the bound sugar.

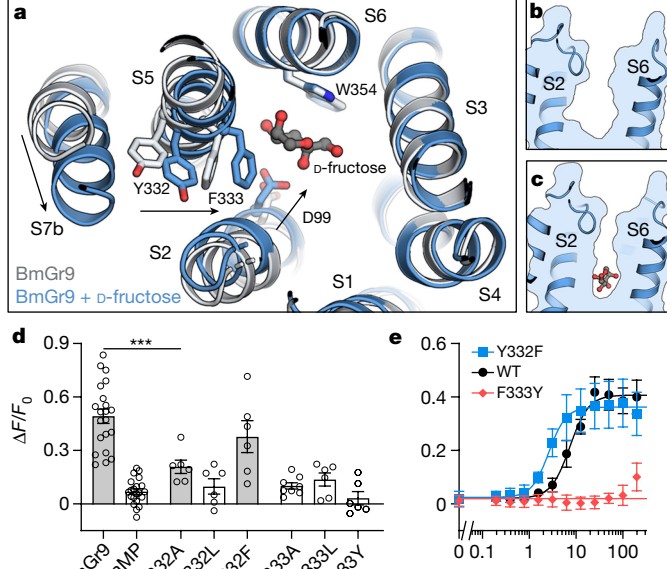

**Fig. 5 | Aromatic bridge connecting D-fructose to the channel pore.**
**a**, Conformational changes following binding of D-fructose. Arrows indicate movement of key regions in BmGr9 from the unbound (grey) to the bound (blue) state. **b,c**, Outline of the sugar-binding pocket in the absence (**b**) or presence (**c**) of D-fructose, showing that the pocket volume decreases following binding of D-fructose. **d**, Fluorescence changes of BmGr9 and aromatic bridge mutants when stimulated with D-fructose. Bars are mean ± s.e.m. with replicates shown as open circles; grey bars indicate statistically significant activity compared to that of GCaMP alone. Statistical significance was determined using one-way ANOVA followed by Dunnett's multiple comparison tests (***$P < 0.0001$) **e**, Dose response of select mutants. Data points are mean ± s.e.m. from $n = 6$ (Y332F and F333Y) or 8 (wild type) independent experiments collected on the same plates.

In the closed state, Tyr332 and Phe333 are exposed to the membrane interior through a gap between the S7b and S2 helices, where additional columnar lipid or detergent density is observed in the structure of unbound BmGr9 (Extended Data Fig. 6e). However, when the pore opens, Leu441 in S7b closely associates with Met91 and Val95 in S2, expelling the putative lipid or detergent and forming a network of interactions that shield the aromatic bridge from the membrane (Extended Data Fig. 6f). Nearby, Arg90 in S2 reaches across the gap between subunits to interact with carboxy-terminal carbonyls of S5 in the absence of ligand. Movement of S2 towards D-fructose breaks this interaction, increasing the space between subunits and accommodating the shift in lipid positions near the aromatic bridge. This transformation of the membrane-facing surface of BmGr9 between the open and closed states of the receptor raises the possibility that the lipid environment will affect the gating properties of the receptor.

## Mechanism of receptor tuning

Although BmGr9 is activated only by D-fructose, whether other sugars can bind to the receptor is a central matter for determining the origin of this selectivity. To identify other sugars that can potentially bind BmGr9, we computationally docked sweet molecules into the fructose-bound structure of BmGr9. We found that most hexoses similar in size to D-fructose fit well into the binding pocket and make many of the same contacts, yielding similar docking scores (Extended Data Fig. 8). However, larger sugars, such as the disaccharide sucrose, are too big to fit and their lowest-energy poses sit outside the pocket.

To determine experimentally whether sugars with favourable docking scores bind BmGr9, we measured intrinsic tryptophan fluorescence of detergent-solubilized BmGr9. The sugar-binding site in BmGr9 contains two Trp residues (Trp193 and Trp354), which become less exposed to water once D-fructose is bound. The Trp fluorescence emission spectrum of BmGr9 has a maximum near 330 nm; adding a saturating amount of D-fructose decreases the emission intensity and blueshifts the spectrum to a maximum at 324 nm (Fig. 6a), consistent with the Trp residues being buried by D-fructose. The apparent affinity of purified BmGr9 for D-fructose determined using this fluorescence-based assay ($K_d = 16$ mM; Fig. 6b) is in close agreement with our measurement in cells (9 mM; see Fig. 1b) and with previous work on BmGr9 (refs. 2,16).

Titration of BmGr9 with other hexoses produces a blueshift in the Trp fluorescence spectrum, similar to that induced by D-fructose (Fig. 6b,c). Consistent with our docking results, our data show that L-sorbose binds BmGr9 with an affinity close to that of D-fructose whereas sucrose did not induce a shift in the tryptophan fluorescence spectrum. As other sugars can bind BmGr9, the precise positioning of amino acids in the pocket is insufficient, by itself, to fully explain the narrow tuning of the receptor.

As other hexoses are predicted to make many of the same contacts as D-fructose, it is unclear why they do not activate BmGr9. D-Fructose and L-sorbose differ by the chirality of a single hydroxyl group at the C5 position. The structure of BmGr9 bound to β-D-fructopyranose reveals that the C5 hydroxyl (C6 hydroxyl in β-D-fructofuranose) forms a hydrogen bond with Gln351, leaving a hydrophobic patch consisting of aliphatic hydrogens from C4, C5 and C6 to face Phe333 (Fig. 2c,d). This hydrophobic interaction between D-fructose and Phe333 seems critical for repositioning the aromatic bridge closer to the bound sugar, thereby creating space for the pore helix to open. In L-sorbose, the inverted stereochemistry of C5 places a hydroxyl group in the middle of the hydrophobic patch, probably impeding proper engagement of the aromatic bridge and preventing binding being transduced to pore opening. To test our hypothesis, we determined the structure of BmGr9 in the presence of a saturating amount of L-sorbose to a nominal resolution of 2.6 Å (Extended Data Fig. 2k–n). L-Sorbose exists predominantly in a six-membered ring conformation (about 98% α-L-sorbopyranose)[17,18], whose single hydroxymethyl arm matches the asymmetric density for the bound ligand well (Fig. 6d and Extended Data Fig. 3g,k). In the pocket, L-sorbose makes many of the same interactions as D-fructose: a hydrophobic surface (hydrogens of C1 and C3) faces Trp354 and the hydroxyl groups are all coordinated (Fig. 6e). However, L-sorbose does not interact with Phe333 of the aromatic bridge; therefore, there is no link between L-sorbose in the binding pocket and the pore helix (Fig. 6f and Extended Data Fig. 6g), leaving the ion-conducting pore in a closed conformation (Extended Data Fig. 7d). This structure, with a bound ligand but closed pore, illustrates how the structural changes associated with ligand binding may be uncoupled from those leading to channel activation.

Our results thus indicate that receptor tuning in BmGr9 arises from two distinct but complementary mechanisms: the stereoelectronic characteristics of the ligand-binding pocket; and the engagement of an allosteric process required to activate the channel. Altering contributors to these two elements results in different effects on BmGr9 activity. Whereas substitutions within the ligand-binding pocket generally abolished channel activity, many amino acid substitutions along the allosteric pathway were not only tolerated but also created channels that were more active than those of the wild-type BmGr9 (that is, Y332F, Q443E and Q445A). Taken together, our data suggest that pocket characteristics have large effects on receptor chemical sensitivity, by modulating which ligands can be recognized, whereas alterations along the allosteric pathway change how binding events are coupled to pore opening, providing a mechanism to tune the selectivity of channels that share similar binding pockets.

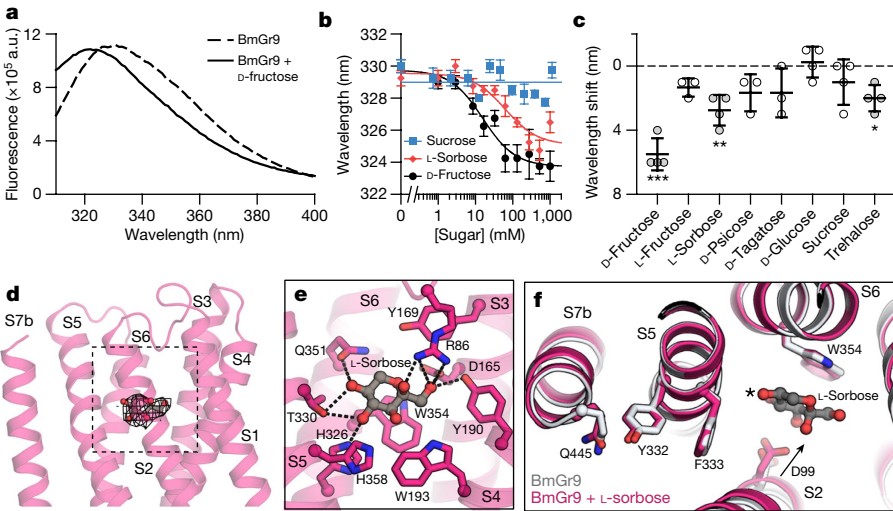

**Fig. 6 | Several sugars bind BmGr9. a**, Representative tryptophan fluorescence emission spectra of BmGr9 in the absence (dashed line) and presence (solid line) of 100 mM D-fructose. a.u., arbitrary units. **b**, Titration of purified BmGr9 with hexoses induces a blueshift in the tryptophan fluorescence emission spectrum whereas that with larger sugars does not. The fitted $K_d$ values for D-fructose and L-sorbose are 15.8 (6.4–38.5) mM and 63 (28–140) mM, respectively ($n = 4$ independently purified samples; points are mean ± s.e.m.; fitted 95% confidence intervals are given in parentheses). **c**, Change in Trp emission maxima in the

presence of 100 mM of a variety of sugars (mean ± s.d., with independent samples shown as open circles). Statistical significance was determined using paired $t$-tests comparing fluorescence before and after sugar addition (***$P = 0.0016$, **$P = 0.010$, *$P = 0.016$). **d**, Structure of BmGr9 bound to α-L-sorbopyranose (grey, carbons; red, oxygens), viewed from the side. **e,f**, Close-up views showing interactions between BmGr9 and α-L-sorbopyranose. Polar interactions are drawn as dashed lines. In **f**, the C5 hydroxyl facing Phe333 is indicated by an asterisk.

## Discussion

Sweet taste receptors serve the essential role of identifying necessary nutrients, while also contributing to the pleasurable perception of consuming sweet foods. We have determined structures of a eukaryotic sweet taste receptor, offering a unique entry point to investigate the biophysical basis for sweet taste and providing a foundation for understanding how closely related sugars may be discriminated.

BmGr9 is part of an ancient and highly conserved subfamily of nutrient sensors that are activated only by D-fructose, and are expressed in the brains and mouthparts of insects[2,3]. Sugar specificity in BmGr9 is partly achieved by the arrangement of specific amino acids in the ligand-binding pocket, which creates a set of interactions that precisely match the overall shape and pattern of chemical groups in D-fructose. Density in the ligand-binding pocket is consistent with multiple forms of D-fructose (β-pyranose and β-furanose) being present and able to activate BmGr9. Despite small structural differences between the two conformers, both ring conformations share a similarly situated hydrophobic patch that interacts with Phe333. This hydrophobic interaction seems critical for repositioning the aromatic bridge (composed of Tyr332 and Phe333) closer to the bound sugar, which allows the pore to open.

Residues that line the pocket are highly conserved among fructose-selective gustatory receptors, but not in other sugar-sensing gustatory receptors, consistent with the chemistry of the ligand-binding pocket shaping sugar specificity. Notably, in structures of BmGr9 bound to D-fructose or L-sorbose, we observed that Trp354 interacts in a similar manner with both sugar molecules, positioning them within the pocket in BmGr9 so that their hydroxyl groups can be coordinated by constellations of polar interactions. As the hydrophobic surfaces of carbohydrates differ, the aromatic–sugar interaction provides an additional mechanism for discriminating between chemically similar molecules. Aromatic residues are often present within the predicted binding pockets of gustatory receptors and olfactory receptors, raising the possibility that they serve a widespread fundamental role in orienting ligands, comparable to other residues that interact with defined chemical groups through ionic or hydrogen bonds[25,26].

Despite the seemingly specific interactions of BmGr9 with D-fructose, our computational docking and experimental binding assays show that other sugars can bind BmGr9. Indeed, our structure of BmGr9 bound to L-sorbose reveals that non-activating sugars not only can fit into the ligand-binding pocket, but also can make many of the same contacts as D-fructose, yet the pore is closed. Thus, receptor–ligand interactions in the pocket cannot explain the selective activation by only D-fructose. Instead, we have identified a central switch, the aromatic bridge, that couples ligand binding to pore opening. In our model for BmGr9 selectivity, only D-fructose can both fit into the pocket and simultaneously induce a conformational change in the aromatic bridge, thereby coupling ligand binding to pore opening. Other molecules, such as L-sorbose, may bind in the pocket but they are not able to shift the aromatic bridge into an open configuration. The challenge of discriminating between molecules that differ only by the relative positions of a few hydroxyl groups might be too great for pocket structure alone, necessitating an additional layer of chemical selection in BmGr9.

Most sugar-sensing gustatory receptors are activated by numerous sweet compounds[6]. BmGr9, therefore, represents an extreme example of specificity within this family. How other gustatory receptors achieve broad selectivity is unclear. One possibility is that their binding pockets can interact with many more sugars than BmGr9. This mechanism would resemble that of MhOr5, in which the generic binding pocket adapts to accommodate differently shaped odorants[14]. Our work suggests an alternative possibility: that many sugars may bind each gustatory receptor, but only some can induce the appropriate conformational changes necessary to reach the activation threshold. A likely scenario is that both mechanisms contribute to defining receptor tuning: coarse receptor tuning is derived from the size and chemical characteristics of the pocket, which restricts the set of molecules that can bind, whereas fine-tuning is achieved through the selective engagement of an allosteric pathway that connects the pocket to the pore. BmGr9 would then occupy one extreme of the tuning spectrum, with broadly tuned chemoreceptors at the other potentially being activated by any molecule able to fit into the ligand-binding pocket.

This two-layer mechanism is probably a general feature of other chemoreceptor families. Inhibition of insect and mammalian olfactory

receptors by odorants is prevalent[27,28], indicating that many molecules bind but do not activate these receptors. This inhibition provides an important feature of the combinatorial coding of odour mixtures, suggesting that the coupling between ligand binding and receptor activation may be a point of evolutionary selection that can tune the activity of receptors with similar binding pockets. Continued investigation into chemoreception by diverse receptors will help us explore the relationship between amino acid sequence, specificity and receptor tuning—ultimately revealing how families of receptors work together to decipher the chemical world.

*Note added in proof:* A related study presenting structures of homologous insect sugar receptors was recently published (ref. 29).

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

# Methods

## Expression and purification of BmGr9

A synthetic construct consisting of residues Pro2–Ser449 (the native C terminus) of BmGr9 (GenBank accession EU769120.1) was cloned into a pEG BacMam vector (Addgene plasmid number 160451; from E. Gouaux)[30] following an amino-terminal Strep-tag II[31], superfolder GFP[32] and an HRV 3C protease site. Baculovirus containing the BmGr9-coding sequence was created in Sf9 cells (ATCC CRL-1711, neither authenticated nor checked for mycoplasma contamination). HEK293S GnTI⁻ cells (ATCC CRL-3022, neither authenticated nor checked for mycoplasma contamination) were grown in suspension at 37 °C in Freestyle 293 medium (Gibco) supplemented with 2% (v/v) fetal bovine serum (FBS; Gibco) and 1% (v/v) GlutaMAX (Gibco) with 8% (v/v) carbon dioxide until they reached a density of about $3 \times 10^6$ cells per millilitre and then transduced with baculovirus at a multiplicity of infection of about 1. After 12 h, 10 mM sodium butyrate (Sigma-Aldrich) was added to the medium, and the temperature was reduced to 30 °C. The cells were collected about 48 h later by centrifugation and washed once in phosphate-buffered saline (pH 7.5; Gibco). Cell pellets were frozen in liquid nitrogen and stored at −80 °C until needed.

Cell pellets were thawed on ice and resuspended in 20 ml of lysis buffer per gram of cells. Lysis buffer was composed of 50 mM HEPES–NaOH (pH 7.5), 375 mM NaCl, 10 μg ml⁻¹ DNase I, 1 μg ml⁻¹ leupeptin, 1 μg ml⁻¹ aprotinin, 1 μg ml⁻¹ pepstatin A and 1 mM phenylmethylsulfonyl fluoride (all from Sigma-Aldrich). BmGr9 was extracted by adding 1% (w/v) n-dodecyl-β-D-maltoside (DDM; Anatrace) with 0.2% (w/v) cholesterol hemisuccinate (CHS; Sigma-Aldrich) for 2 h at 4 °C. The mixture was clarified by centrifugation at 80,000g and the supernatant was added to 0.4 ml StrepTactin Sepharose resin (Cytiva) per gram of cells and rotated at 4 °C for 1 h. The resin was collected, washed with 10 column volumes of 20 mM HEPES–NaOH (pH 7.5) containing 150 mM NaCl (HEPES-buffered saline (HBS)) with 0.02% (w/v) DDM and 0.004% (w/v) CHS, and then with 10 column volumes of HBS with 0.05% (w/v) digitonin (Sigma-Aldrich). BmGr9 was eluted by adding 2.5 mM desthiobiotin (Sigma-Aldrich) to the digitonin buffer.

The StrepII–GFP tag was cleaved by HRV 3C protease (Novagen) added at 10 U mg⁻¹ of BmGr9 overnight at 4 °C. BmGr9 concentration was calculated from its absorbance at 280 nm assuming an extinction coefficient ($\varepsilon_{280}$) of 45.8 mM⁻¹ cm⁻¹ (calculated by ProtParam[33]). BmGr9 was then concentrated in a centrifugal tube (Amicon Ultra-4; 100-kDa cutoff) and injected onto a Superose 6 Increase column (Cytiva) previously equilibrated with HBS with 0.05% (w/v) digitonin. For the sugar-bound samples, 0.5 M D-fructose (Sigma-Aldrich) or 1 M L-sorbose (Sigma-Aldrich) was included in the buffer.

## Cryogenic electron microscopy sample preparation and data collection

Peak fractions containing purified BmGr9 were concentrated to 4.7 mg ml⁻¹ (unbound sample), 2.9 mg ml⁻¹ (fructose-bound sample) or 4.6 mg ml⁻¹ (sorbose-bound sample). Cryogenic electron microscopy (cryo-EM) grids were frozen using a Vitrobot Mark IV (FEI) as follows: 3 μl of the concentrated sample was applied to a glow-discharged Quantifoil R1.2/1.3 holey carbon 400 mesh gold grid, blotted for 2.5–4 s in >90% humidity at room temperature, and plunge frozen in liquid ethane cooled by liquid nitrogen. Grids were screened for ice thickness and particle distribution using a Glacios (200 kV; Thermo Scientific) in the Yale School of Medicine Center for Cellular and Molecular Imaging.

Cryo-EM data were recorded on a Titan Krios (300 kV; FEI) in the Yale West Campus Cryo-EM Core, equipped with a Gatan K3 Summit camera and imaging filter. SerialEM (version 4.1-beta)[34] was used for automated data collection. Videos were collected at a nominal magnification of ×81,000 in super-resolution mode resulting in a calibrated pixel size of 0.534 Å per pixel, with a defocus range of approximately −0.8 to −2.5 μm. Fifty frames were recorded over 10 s of exposure at a dose rate of 1.67 electrons per square ångström per frame. Video frames were aligned and binned over 2 × 2 pixels using MotionCor2 (ref. 35) and the contrast transfer function parameters for each motion-corrected image were estimated using CTFFIND4.1 (ref. 36).

An initial set of about 2,000 BmGr9 particles were manually picked and submitted to two-dimensional (2D) class average to create references for auto-picking. For the unbound BmGr9 sample, 1,179,559 particles from 5,770 micrographs were extracted, into 384 × 384-pixel boxes, binned over 3 × 3 pixels, and subjected to further 2D classification using RELION-3.1 (ref. 37). After removing junk particles, the remaining 446,673 particles were re-extracted without binning and used to build an initial 3D model in RELION-3.1. The model was further refined with $C_4$ symmetry imposed, ultimately reaching 3.8 Å resolution without masking. The particles were then imported into CryoSPARC (version 3.3.1, Structura Biotechnology)[38] and underwent a new round of 2D classification. After carrying out subsequent rounds of non-uniform refinement and local refinement (with $C_4$ symmetry), local contrast transfer function refinement and further 2D classification, we obtained a map with a nominal resolution of 2.9 Å, estimated using the Fourier shell correlation = 0.143 cutoff criterion[39].

A similar approach was used for the fructose-bound BmGr9. A total of 1,760,600 particles from 7,035 micrographs were auto-picked in RELION-3.1. After initial 2D classification, 1,157,505 particles were extracted and moved to CryoSPARC for further processing. We used 307,715 particles to build the final map with a 3.0 Å resolution. For the L-sorbose-bound sample, we used CryoSPARC Live to select 13,477 micrographs; 920,021 particles were used in the final refinement, yielding a map with 2.6 Å resolution. The density images in Fig. 1c were created using UCSF ChimeraX (version 1.7)[40].

## Cryo-EM data analysis and model building

Both maps were of sufficient quality for de novo atomic model building. A poly-alanine model for BmGr9 was built in Coot (version 0.9.8.1)[41], and subsequent amino acid assignments were made on the basis of side-chain densities. The models were refined using real-space refinement implemented in PHENIX[42] for five macro-cycles with four-fold non-crystallographic symmetry and secondary structure restraints applied. The lowest-scoring docking poses (see below) were used as the starting positions to refine the structures of D-fructose conformers. Ligand restraints were obtained using the electronic Ligand Builder and Optimization Workbench[43] implemented in PHENIX (version 1.20.1-4487).

There was no apparent density for the intracellular loop between S4 and S5 in any of our samples; hence, all final models lack this region. Model statistics were obtained using MolProbity[44]. Models were validated by randomly displacing the atoms in the original model by 0.5 Å, and refining the resulting model against half-maps and the full map. Images of the model were created using PyMOL (version 2.5)[45] and UCSF ChimeraX. Fischer and Haworth projections of sugars were made using ChemDraw 22.2 (PerkinElmer).

## Structural analysis

Residues at subunit interfaces or the binding pocket were identified using PyMOL as any residue within 5 Å of a neighbouring subunit or the ligand, respectively. The pore diameters along the central axis and lateral conduits (in Fig. 4a–d) were calculated using the program HOLE (64-bit Linux version v2.2.005)[46]. Two calculations were carried out: one along the central four-fold axis (central pore) and another between subunits near the cytosolic membrane interface (lateral conduits). The pores overlapped in the central vestibule.

## Computational docking

Molecular docking was carried out with the apo and fructose-bound models and a ligand set of various sweet compounds using the molecular docking software AutoDock Vina (version 1.2.0)[19,20] and the Vinardo[47]

scoring function. As some carbohydrates assume different conformations in solution, all anomeric forms were considered. Cubical grids of different sizes and locations in and around the protein were generated using the grid feature of AutoDock Tools to determine the best docking space within the protein. In the end, a 4,500-$Å^3$ cubical grid was centred in the observed binding pocket with $x$- and $y$-axis lengths of 15 Å and a $z$ axis length of 20 Å. The structures were prepared in AutoDock Tools by adding any missing atoms and charges on residues assuming a pH of 7.4. Compounds structures (SDF files) were downloaded from the Research Collaboratory for Structural Bioinformatics PDB, PubChem and ZINC databases and prepared for docking by conversion into PDBQT files using OpenBabel (version 3.1.1)[48].

## Cell-based GCaMP calcium flux assay

The cell-based GCaMP assay was based on a previously described method used to study insect olfactory receptors[13]. BmGr9 variants were cloned into a modified pME18s vector with no fluorescent tag, flanked by AscI/NotI restriction enzyme sites for efficient cloning. Each transfection condition contained 0.5 µg of a plasmid encoding GCaMP6s (Addgene plasmid number 40753; from D. Kim & GENIE Project) and 1.5 µg of the plasmid encoding the BmGr9 variant, in 250 µl OptiMEM (Gibco). These were mixed with a solution containing 7 µl Lipofectamine 3000 (Invitrogen) in 250 µl OptiMEM, followed by a 15-min incubation at room temperature.

HEK293 cells (ATCC CRL-1573, neither authenticated nor checked for mycoplasma contamination) were maintained in high-glucose DMEM supplemented with 10% (v/v) FBS and 1% (v/v) GlutaMAX with 5% (v/v) carbon dioxide at 37 °C. Cells were detached using trypsin and resuspended to a final concentration of $1 \times 10^6$ cells per millilitre. Cells were mixed with the transfection mixture and added to a 384-well plate with a clear bottom (Grenier). Four to six hours later, a 16-port vacuum manifold on low vacuum was used to remove the transfection medium, which was replaced by fresh FluoroBrite DMEM (Gibco) supplemented with 10% (v/v) FBS and 1% (v/v) GlutaMAX. Twenty-four hours later, this medium was replaced with 50 µl reading buffer (20 mM HEPES–NaOH (pH 7.4), 0.1× Hank's balanced salt solution (Gibco), 3 mM $Na_2CO_3$ and 5 mM $CaCl_2$) in each well.

The fluorescence emission at 515–575 nm, with excitation at 475–495 nm, was continuously read by a FLIPR Tetra System (Molecular Devices) at the Yale Center for Molecular Discovery. The exposure time was set to 0.5 s, excitation intensity was set at 100%, and camera gain was adjusted according to the baseline signal for each plate. After 30 s of baseline recording, 25 µl of tastant solution was added to the cells and read for 8 min. All solutions were pre-warmed to 37 °C. All sweet compound titrations were made using 11 ligand concentrations for each transfection condition in sequential dilutions of 2, alongside control wells of reading buffer alone. Ligands were dissolved in the reading buffer at 600 mM, and then diluted with reading buffer to the highest final concentration of 200 mM. Owing to their lower solubility, the sweeteners aspartame and saccharin had a highest final concentration of 10 mM. Ligand concentrations for mutants were the same as for the wild type.

Each plate contained a negative control of GCaMP6s transfected alone and exposed to tastants. Additionally, each plate included BmGr9 with its cognate ligand D-fructose as a positive control for plate-to-plate variation in transfection efficiency and cell count. Each ligand concentration was applied to two technical replicates, which were averaged and considered a single biological replicate. The baseline fluorescence ($F_0$) was calculated as the average fluorescence of the 30 s before any addition to the plate. The maximum signal was reached 50–70 s after tastant addition, and the average fluorescence signal in that period ($F$) was used for further calculations. $F - F_0/F_0$ for each concentration was calculated to account for well-to-well variability. In the sugar panel, the percentage of activity was calculated as the difference in fluorescence change evoked by adding 100 mM of sweet compound (or 10 mM of

aspartame and saccharin) between cells transfected with BmGr9 plus GCaMP and control cells transfected with only GCaMP, divided by the difference observed when treated with D-fructose (multiplied by 100 to yield a percentage). For all experiments, GraphPad Prism 10 was used to fit all dose–response curves to the four-parameter Hill equation, from which $EC_{50}$ and Hill coefficients were extracted.

## Expression tests

HEK293 cells were maintained in high-glucose DMEM supplemented with 10% (v/v) FBS and 1% (v/v) GlutaMAX (all from Gibco) at 37 °C with 5% (v/v) carbon dioxide. Cells were detached using 1× trypsin (Gibco) and seeded in six-well plates at a concentration of $1 \times 10^6$ cells per well. Cells in each well were transfected approximately 24 h later with 3 µg of DNA (GFP-tagged BmGr9 variants in the same pEG BacMam vector used for large-scale purification) and 10 µl Lipofectamine 3000 (Invitrogen), diluted in 500 µl OptiMEM (Gibco) and added dropwise after a 20-min incubation. Cells were checked for GFP fluorescence 24 h later, rinsed and resuspended with phosphate-buffered saline (Sigma-Aldrich), and collected by centrifugation at 20,000$g$ for 5 min. Cells pellets were used immediately and resuspended in 200 µl lysis buffer containing 50 mM HEPES–NaOH (pH 7.4), 375 mM NaCl, 10 µg $ml^{-1}$ DNAse I, 1 µg $ml^{-1}$ leupeptin, 1 µg /$ml^{-1}$ aproptonin, 1 µg $ml^{-1}$ pepstatin A and 1 mM phenylmethylsulfonyl fluoride. The protein was extracted for 2 h at 4 °C by adding 1% (w/v) DDM with 0.2% (w/v) CHS after 10 s sonication in a water bath. This mixture was then clarified by centrifugation at 19,000$g$ for 30 min and filtered. An aliquot of the supernatant was used to run SDS–PAGE (Bio-Rad, 12% Mini-PROTEAN TGX) and native-PAGE (Invitrogen, 3–12% Bis-Tris) gels. Gels were transferred using a Trans-Blot Turbo Transfer Pack (Bio-Rad) and blocked overnight. The following day, the SDS–PAGE gels were stained with rabbit anti-GFP polyclonal antibody, Alexa Fluor 488 (ThermoFisher, 1:4,000), washed and imaged with ImageLab (Bio-Rad). The native-PAGE gels were incubated with mouse anti-Strep-tag II monoclonal antibody (ThermoFisher, 1:3,000), washed and incubated with anti-mouse HRP secondary antibody (1:10,000).

## Tryptophan fluorescence measurements

BmGr9 was expressed and purified as described above. Sugar solutions were made in the same buffer used for the Superose 6 column. A 100 µl volume of BmGr9 (at 1 µM) was added to a quartz cuvette, and the tryptophan fluorescence was measured in a PTI fluorometer. The excitation wavelength was set to 295 nm with a bandwidth of 20 nm. Fluorescence emission was measured at 300 to 400 nm using the Felix software (version 1.42b, PTI), with a 1-nm step size. After the addition of each ligand, solutions were mixed gently using a pipette, followed by fluorescence measurements after 1 min. All curves were corrected for dilution. The same ligand concentrations were added to 25 µM $n$-acetyl-L-tryptophanamide (NATA) in 0.05% digitonin to account for potential compound inner filter effects. The observed fluorescence quenching in the NATA experiments was subtracted from BmGr9 measurements for every ligand. An independently purified BmGr9 sample was used for each biological replicate. All data were collected and moved to Excel for further analysis and curve normalization. Images were created using GraphPad Prism 10.

## Sequence alignments and logos

All sequence alignments were visualized and plotted using JalView[49]. We used the open-source web-based logo generation program LogOddsLogo (NCBI). All protein sequences were obtained in the UniProtKB or GenBank databases.

## AlphaFold tetramer and structural comparison

We used ColabFold[50] to predict the structure of the BmGr9 homotetramer. Four identical primary sequences of BmGr9 were added to the AlphaFold2 prediction tool in ChimeraX and submitted with default

parameters to Google Colab servers. Images showing model comparisons were generated using PyMOL and UCSF ChimeraX.

## Reporting summary

Further information on research design is available in the Nature Portfolio Reporting Summary linked to this article.

## Data availability

The final cryo-EM maps have been deposited in the Electron Microscopy Data Bank under the accession numbers EMD-42629 (bound to D-fructose), EMD-42628 (unbound) and EMD-43548 (bound to L-sorbose). The final models have been deposited in the PDB under accession numbers 8UVU (bound to β-D-fructopyranose and β-D-fructofuranose, at 75% and 25% occupancy, respectively), 8UVT (unbound) and 8VV3 (bound to α-L-sorbopyranose). Coordinates for Orco and MhOr5, used for structural comparisons in this paper, were obtained from the PDB under accession numbers 6C70 (Orco), 7LIC (MhOr5) and 7LID (MhOr5 bound to eugenol). Source activity and binding data are provided with this paper. For all other data requests, contact J.A.B.

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

**Acknowledgements** We thank J. Carlson, M. Lemmon, B. Noro, V. Ruta and members of the laboratory of J.A.B. for comments on this manuscript and advice throughout the course of this project; K. Yedlin for carrying out early expression tests of BmGr9; M. Llaguno and S. Wu for support with cryo-EM grid screening and data collection; B. Evans for assistance with cryo-EM data analysis software; and L. Abriola for activity measurements. J.V.G. was supported by the Brazilian Federal Agency for Support and Evaluation of Graduate Education (CAPES). This work was financially supported by grants from the Bill and Melinda Gates Foundation, the Whitehall Foundation, and the NIGMS (RM1GM149406) (all to J.A.B.).

**Author contributions** J.V.G. purified BmGr9, collected and analysed cryo-EM data, and carried out GCaMP activity measurements. S.S.-B. carried out docking computations. J.V.G., C.C.C. and M.S. measured tryptophan fluorescence. M.C. carried out the initial screening of sugar-sensing gustatory receptors and preliminary expression tests. C.C.C. and M.S. measured receptor expression levels. J.A.B. supervised all aspects of the research. J.V.G. and J.A.B. wrote the manuscript with input from all authors.

**Competing interests** The authors declare no competing interests.

**Additional information**
**Correspondence and requests for materials** should be addressed to Joel A. Butterwick.

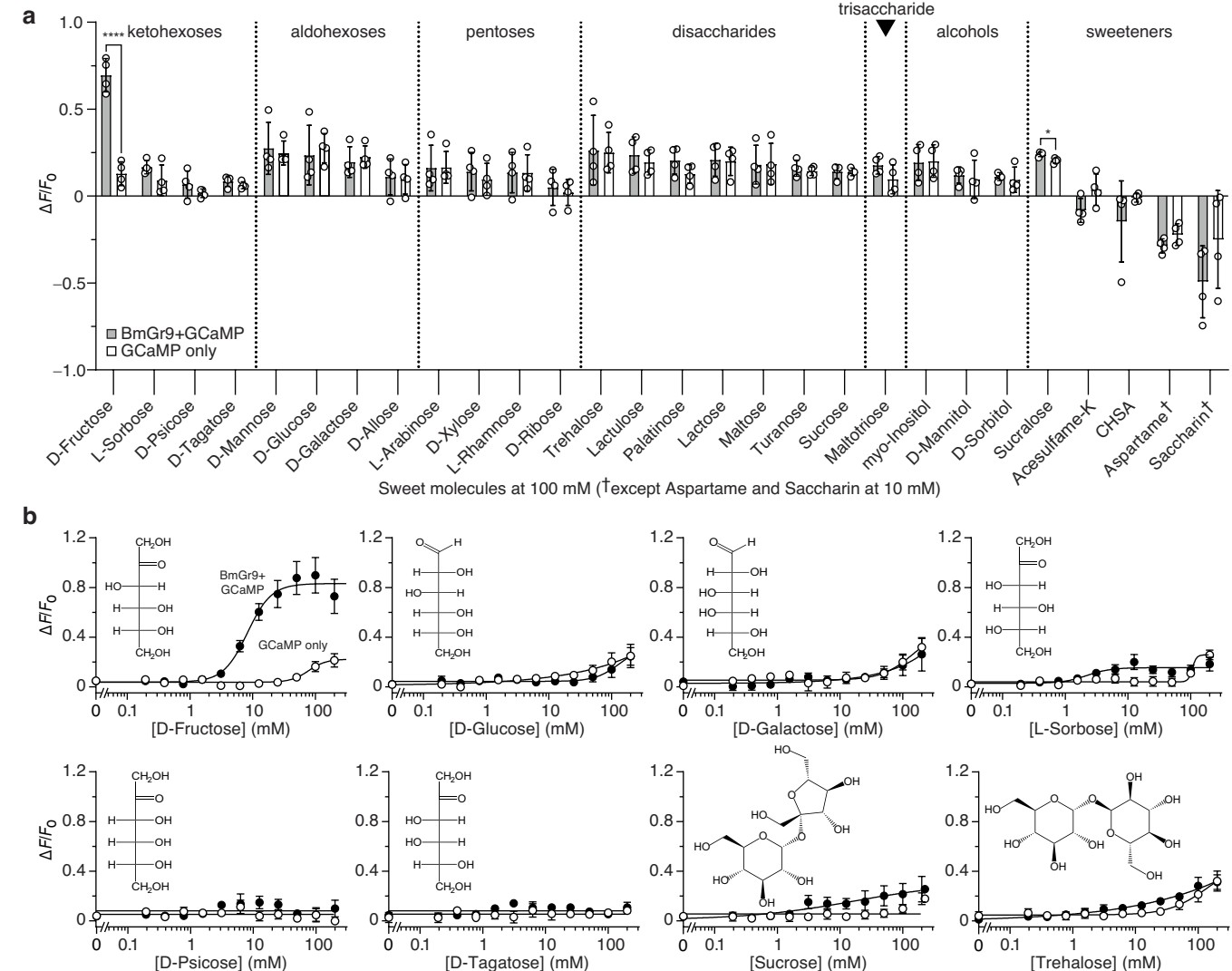

**Extended Data Fig. 1 | BmGr9 is narrowly tuned to D-fructose. a**, Change in fluorescence upon the addition of 100 mM sweet compounds (except for aspartame and saccharin, which were at 10 mM) in HEK293 cells transfected with BmGr9 plus GCaMP (grey bars) or with GCaMP only (white bars). Bars are mean ± s.e.m with replicates ($n = 4$) shown as open circles. Only D-fructose (****$P < 0.0001$) and sucralose (*$P = 0.012$) additions yielded significantly different activity with BmGr9. Statistical significance determined using unpaired t-tests comparing BmGr9 sugar response to respective GCaMP-only controls. **b**, Dose-response of fluorescence changes of HEK293 cells transfected with BmGr9 and GCaMP (closed circles) or GCaMP only (open circles) when titrated with select sugars from (**a**) ($n = 4$, points are mean ± s.e.m.). Insets show Fischer projections of hexoses and perspective projections of myo-inositol, sucrose, and trehalose. For all data, $n$ values represent biological replicates.

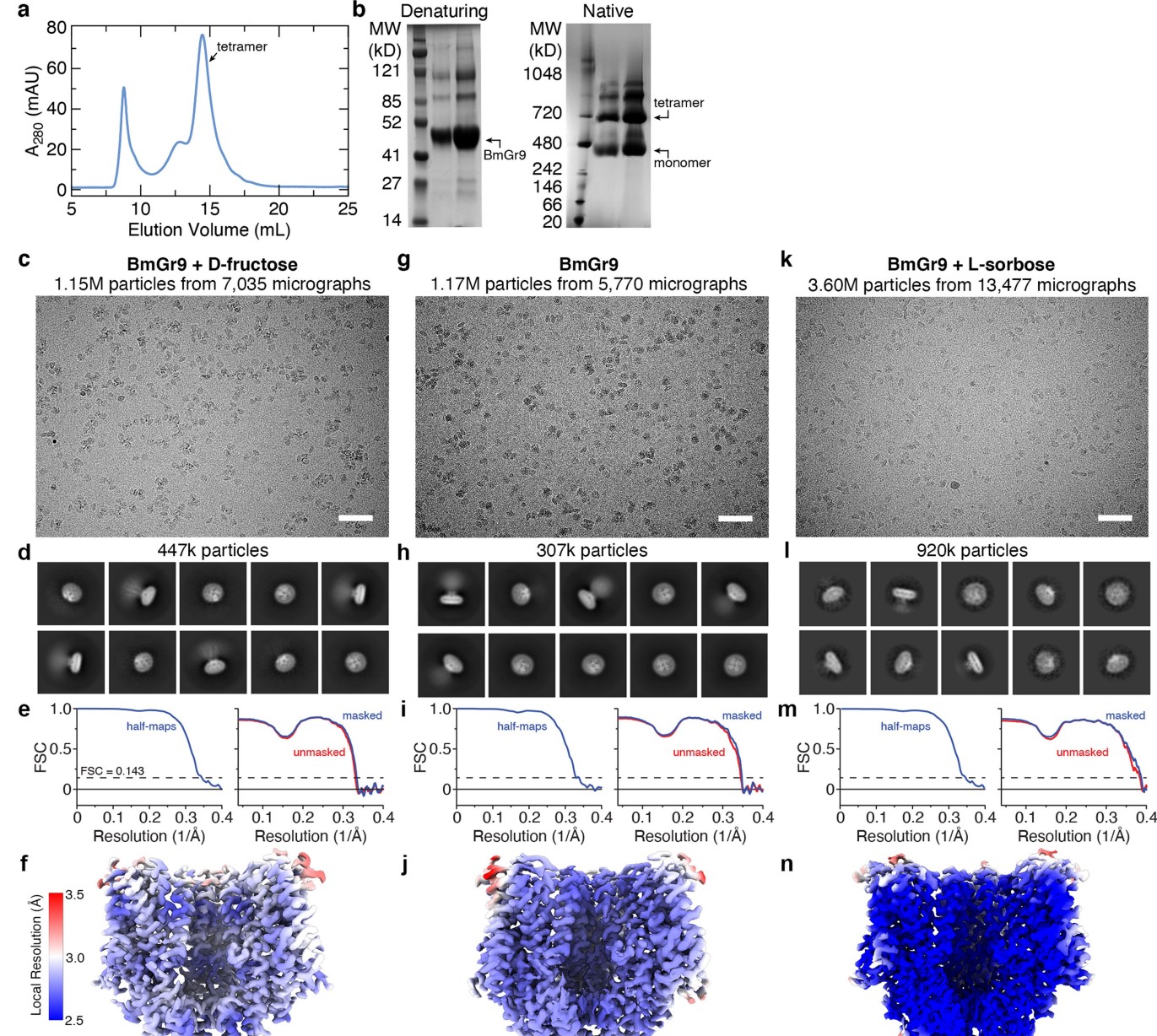

**Extended Data Fig. 2 | Purification of BmGr9 and cryo-EM workflows.**
**a**, Superose 6 elution profile of purified BmGr9. The majority of the protein elutes as a tetramer. **b**, Coomassie staining of denaturing and native gels confirm BmGr9 is a homotetramer with an effective molecular weight of approximately 700 kDa (including detergent micelle), similar to Orco[13]. Molecular weight markers are labeled for each gel (similar results were obtained from more than three independent purifications). **c**, A representative motion-corrected micrograph showing the distribution of fructose-bound BmGr9 single particles (scale bar, 50 nm). The numbers of micrographs and auto-picked particles are

shown. **d**, Example two-dimensional class averages of particles selected for further processing. **e**, Fourier shell correlation (FSC) curves for the final cryo-EM density maps. Half-map FSC (with tight mask) (left), model-map FSC curves (right). The horizontal dashed line represents the FSC = 0.143 cutoff value. **f**, Local resolution of fructose-bound BmGr9 density map viewed from the top (top) and side (bottom). In side views, the nearest subunit has been removed to expose the pore. **g-m**, Equivalent data for unbound BmGr9 (**g-j**) and sorbose-bound Bmgr9 (**k-n**).

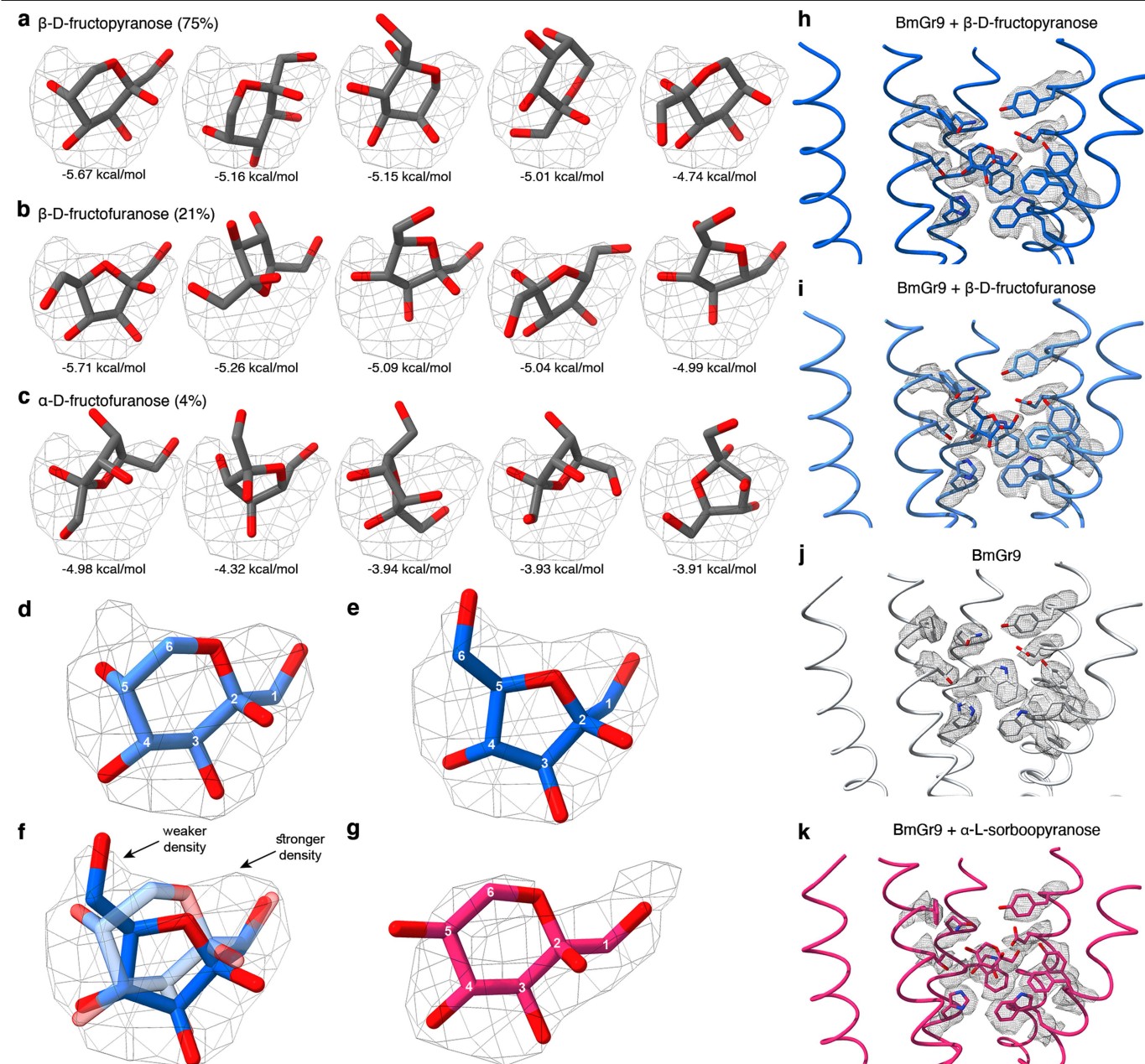

**a** β-D-fructopyranose (75%)

−5.67 kcal/mol    −5.16 kcal/mol    −5.15 kcal/mol    −5.01 kcal/mol    −4.74 kcal/mol

**b** β-D-fructofuranose (21%)

−5.71 kcal/mol    −5.26 kcal/mol    −5.09 kcal/mol    −5.04 kcal/mol    −4.99 kcal/mol

**c** α-D-fructofuranose (4%)

−4.98 kcal/mol    −4.32 kcal/mol    −3.94 kcal/mol    −3.93 kcal/mol    −3.91 kcal/mol

**d**

**e**

**f** weaker density / stronger density

**g**

**h** BmGr9 + β-D-fructopyranose

**i** BmGr9 + β-D-fructofuranose

**j** BmGr9

**k** BmGr9 + α-L-sorboopyranose

**Extended Data Fig. 3 | Computational docking of D-fructose anomers and density within ligand-binding pockets. a-c**, Five lowest energy poses for β-D-fructopyranose, (**a**), β-D-fructofuranose (**b**), and α-D-fructofuranose (**c**) with their respective energy scores (kcal/mol). Experimental ligand density is shown as a grey mesh. The equilibrium composition in solution at room temperature is shown in parenthesis[17,18]. **d,e**, The final positions of β-D-fructopyranose (**d**, light blue) and β-D-fructofuranose (**e**, dark blue) after real-space refinement, with carbons numbered. **f**, Superposition of refined β-D-fructopyranose and β-D-fructofuranose positions. Both ring conformations have similarly positioned hydrogen-bonding groups and hydrophobic surfaces. Stronger density is observed on the side where both conformers of D-fructose have hydroxymethyl groups, consistent with both conformers being bound in our structure and contributing to ligand density. **g**, Final position of α-L-sorbopyranose after real-space refinement, with carbons numbered. **h-k**, Side views of ligand-binding pockets of BmGr9 bound to β-D-fructopyranose (**h**), bound to β-D-fructofuranose (**i**), unbound (**j**), and bound to α-L-sorbopyranose (**k**). Density is shown in light grey.

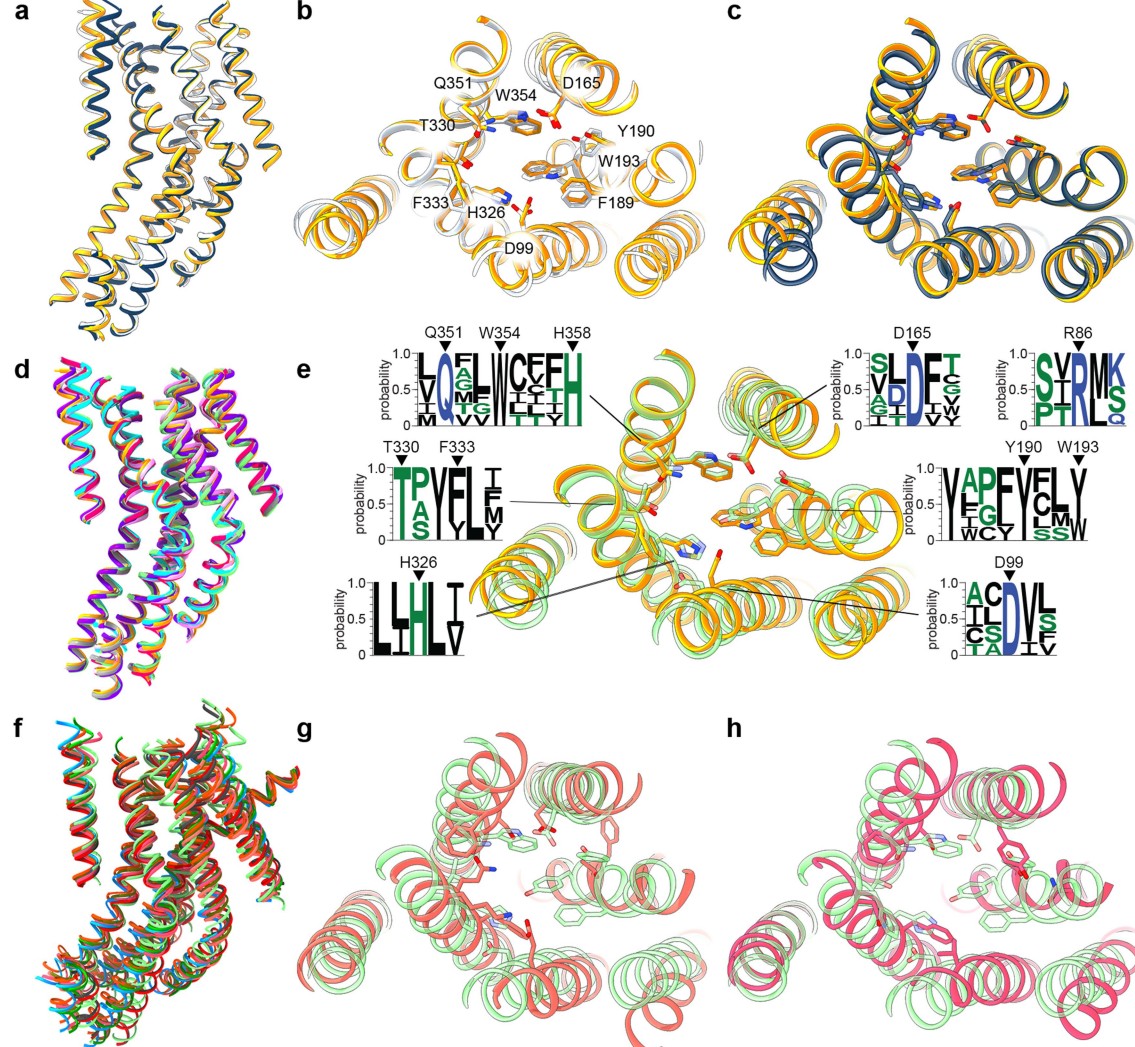

**Extended Data Fig. 4 | Structure conservation among sugar-sensing GRs.**
**a-c**, Superposition of fructose-bound BmGr9 (blue), unbound BmGr9 (white),
and BmGr9 predicted using AlphaFold2 (yellow). Top views (**b,c**) of the ligand-
binding pocket with residues shown (and labelled in (**b**)). **d**, Aligned AlphaFold2
models of known Gr43a-like receptors: BmGr9 (yellow); BmGr10 (purple);
DmGr43a (light green); *Anopheles gambiae* Gr25 (cyan); *Helicoverpa armigera*
Gr9 (grey); *Apis mellifera* Gr3 (light pink); and *Trichogramma chillonis* Gr43a
(dark pink). **e**, Comparison of AlphaFold2 models of BmGr9 and DmGr43a,
with pocket residues shown. Logo representation of amino acid conservation

among the Gr43a-like receptors in (**d**). Residues that interact with D-fructose
are highlighted. **f**, AlphaFold2 models of *D. melanogaster* sugar GRs: DmGr5a
(light red); DmGr43a (light green); DmGr61a (light blue); DmGr64a (magenta);
DmGr64b (dark green); DmGr64c (red); DmGr64d (dark grey); DmGr64e
(orange); and DmGr64f (gold). **g,h**, Comparison of AlphaFold2 models of
DmGr43a (light green) and DmGr5a (**g**, light red) or DmGr64a (**h**, magenta) with
pocket residues shown. No residues in the pocket are conserved between these
receptors. In all images, loops are hidden for clarity.

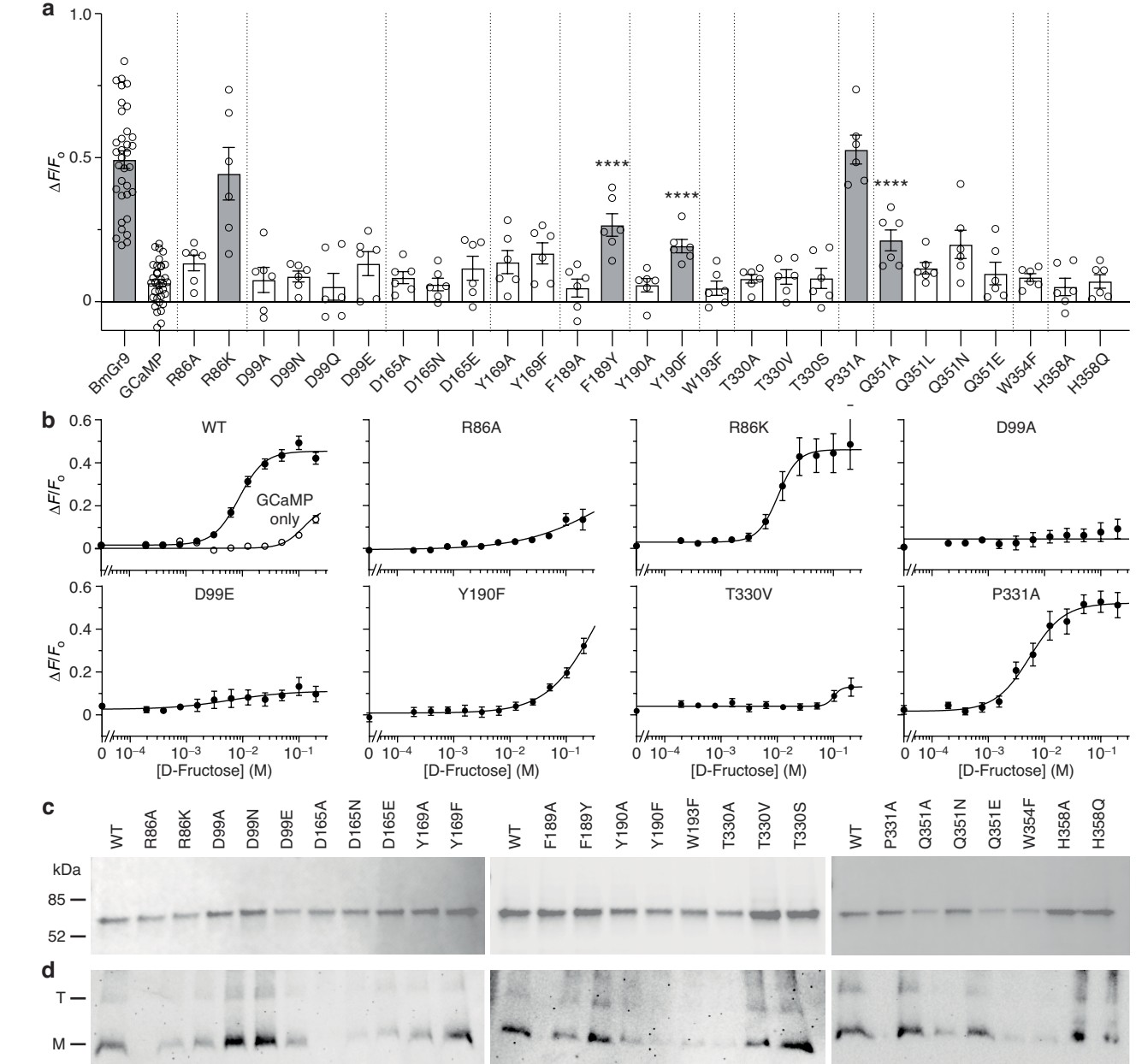

**Extended Data Fig. 5 | Mutational analysis of the sugar-binding pocket in BmGr9. a**, Fluorescence changes of BmGr9 and mutants when stimulated with 100 mM D-fructose. Bars are mean ± s.e.m with replicates shown as open circles; grey bars indicate statistically significant activity compared to GCaMP only. Statistical significance was determined using one-way ANOVA followed by Dunnett's multiple comparison tests. F189Y, Y190F, and Q351A are active, but with significantly decreased activity compared to wild-type (WT) BmGr9

(****$P < 0.0001$). **b**, D-Fructose dose-response curves of HEK293 cells transfected with WT BmGr9 plus GCaMP, GCaMP only, or select mutants with GCaMP. Data points are mean ± s.e.m from $n = 6$ (mutants) or 36 (WT and GCaMP only) independent experiments. **c,d**, SDS-PAGE (**c**) and NativePAGE (**d**) gels showing expression of BmGr9 and mutant receptors (single experiment). Molecular weight markers and position of the monomer (M) or tetramer (T) are indicated. Full gel images are presented in Supplementary Fig. 1.

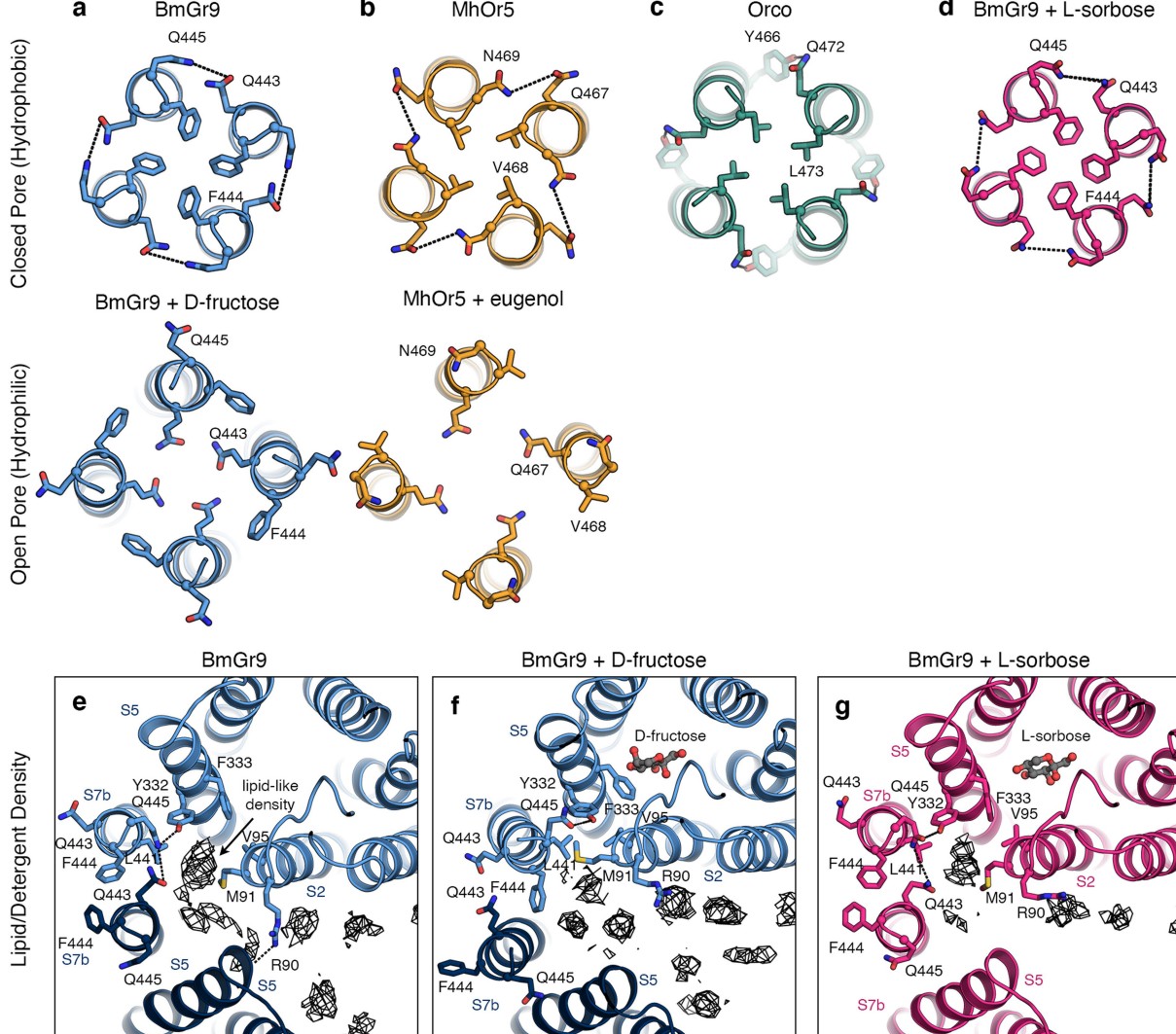

**Extended Data Fig. 6 | Conserved gating in insect chemoreceptors.**
**a-c**, Comparison of pore helices in BmGr9 (**a**), MhOr5 (**b**), and Orco (**c**) in the closed (top) and open (bottom) configurations. In the closed states, hydrophobic residues line the channel gates, which are replaced by hydrophilic residues in the open states. Intersubunit polar interactions are shown as dashed lines. PDB codes are: 7LIC (MhOr5), 7LID (MhOr5 bound to eugenol), and 6C70 (Orco).

**d**, When bound to L-sorbose, BmGr9 maintains a closed conformation.
**e**,**f**, Locations of columnar detergent/lipid-like density (wire mesh) reorganize between the closed (**e**, unbound) and open (**f**, fructose-bound) conformations of BmGr9. The aromatic bridge is only exposed to the lipid environment in the closed state. **g**, Detergent/lipid-like density surrounding BmGr9 when bound to L-sorbose resembles the unbound state.

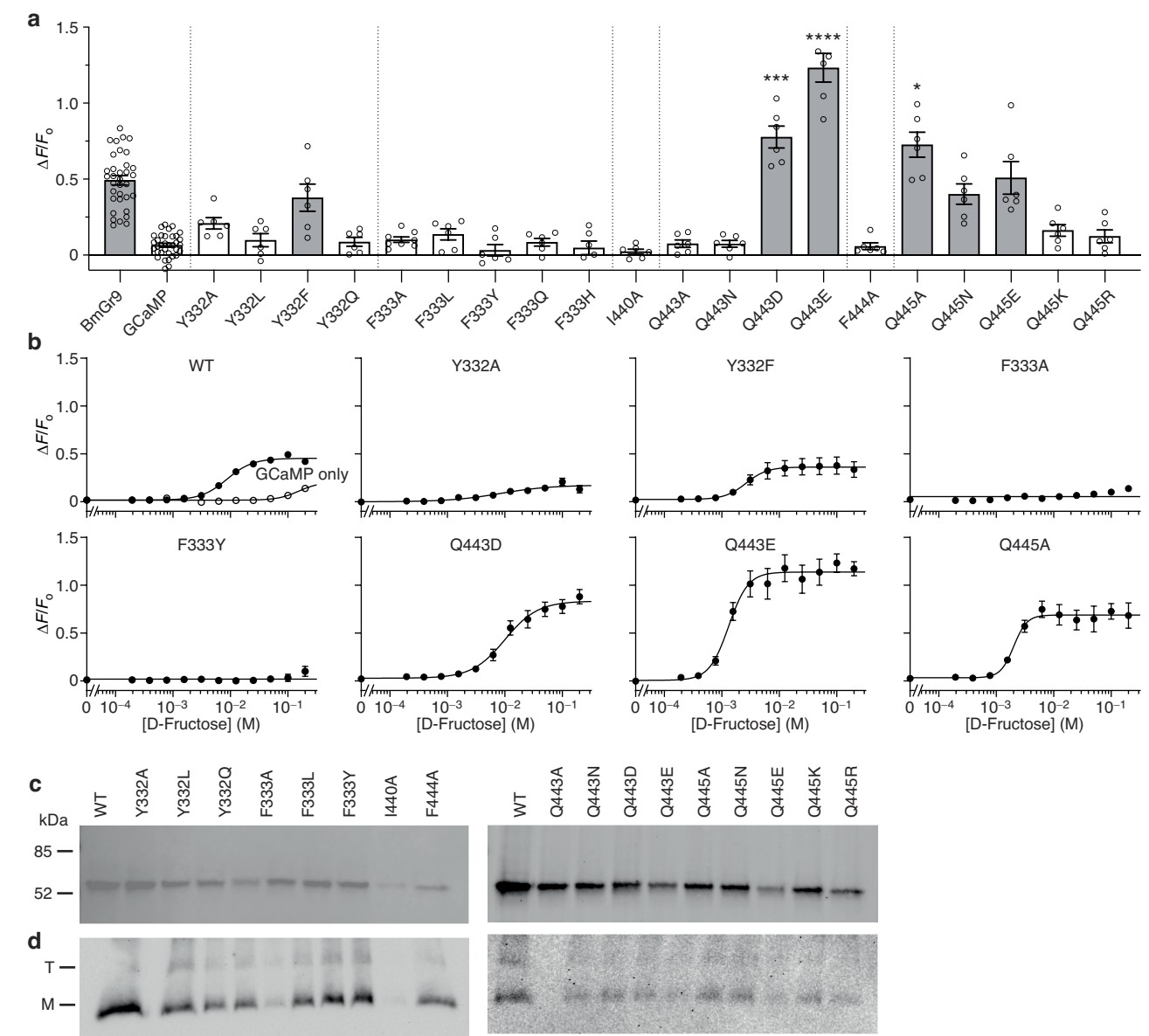

**Extended Data Fig. 7 | Mutational analysis of BmGr9 gating. a**, Fluorescence changes of BmGr9 and mutants when stimulated with 100 mM D-fructose. Bars are mean ± s.e.m with independent replicates shown as open circles; grey bars indicate statistically significant activity compared to GCaMP only. Statistical significance was determined using one-way ANOVA followed by Dunnett's multiple comparison tests. Q443D, Q443E, and Q445A mutation have significantly increased activity compared to wild-type (WT) BmGr9 (\*\*\*\*$P < 0.0001$, \*\*\*$P = 0.0006$, \*$P = 0.012$). **b**, D-Fructose dose-response curves of HEK293 cells transfected with WT BmGr9 plus GCaMP, GCaMP only, or select mutants with GCaMP. Data points are mean points are mean ± s.e.m from $n = 6$ (mutants) or 36 (WT and GCaMP only) independent experiments. WT and GCaMP-only data are the same as presented in Extended Data Fig. 6, but with y-axis adjusted. **c**,**d**, SDS-PAGE (**c**) and NativePAGE (**d**) gels showing expression of BmGr9 and mutant receptors (single experiment). Molecular weight markers and position of the monomer (M) or tetramer (T) are indicated. Full gel images are presented in Supplementary Fig. 1.

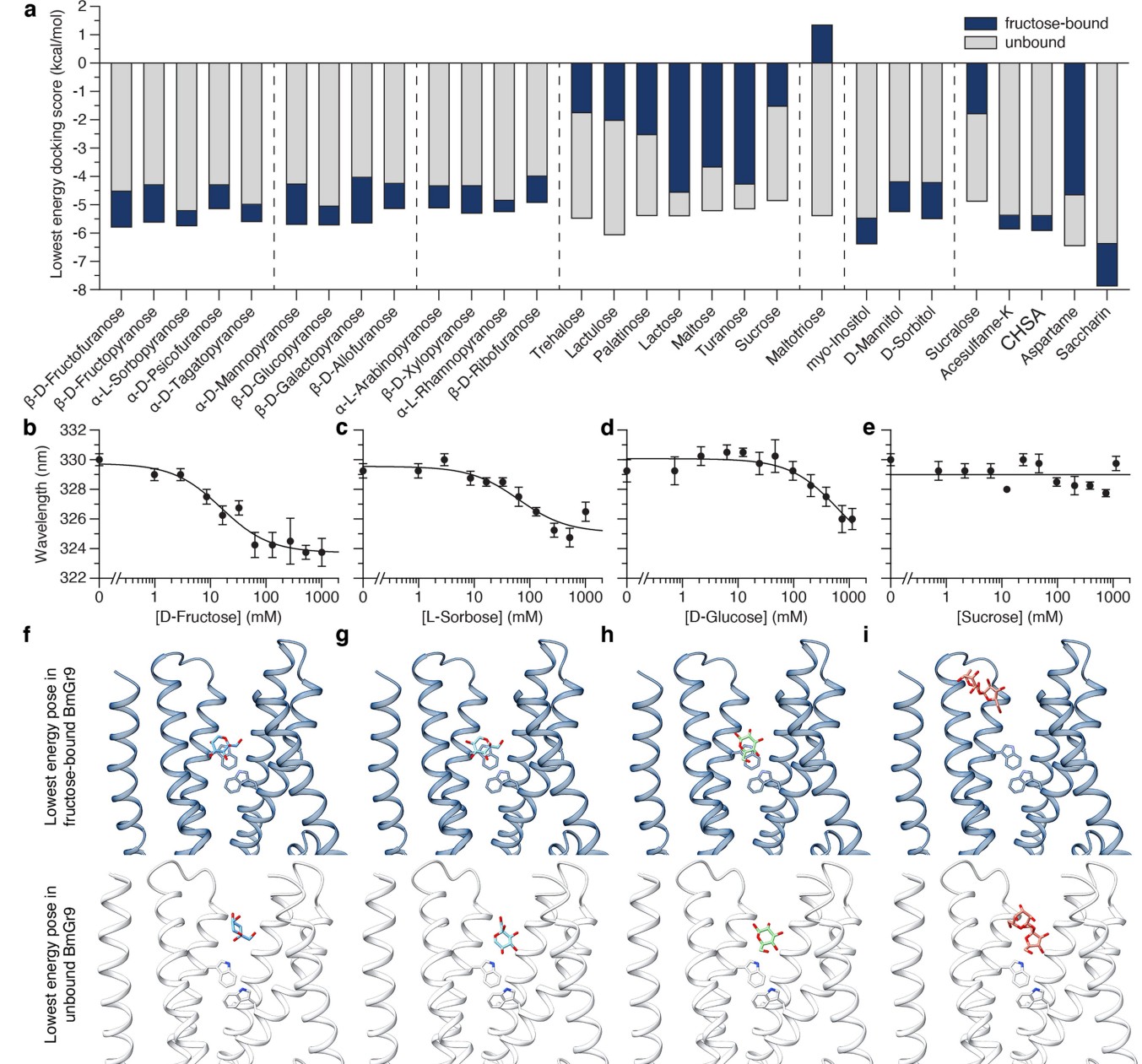

**Extended Data Fig. 8 | Other sweet molecules bind to BmGr9.**
**a**, Superimposed bars of the docking scores (kcal/mol) of sweet molecules in fructose-bound BmGr9 (blue) and apo-BmGr9 (grey). Only predominant anomers in solution were selected for docking, except for D-fructose. **b-e**, Wavelength of the maximum tryptophan fluorescence emission spectrum of BmGr9 when titrated with D-fructose (**b**), L-sorbose (**c**), D-glucose (**d**), and sucrose (**e**). Data points are mean ± s.e.m from $n$ = 4 independently purified samples. **f-i**, Lowest energy poses for docked β-D-fructopyranose (**f**), α-L-sorbopyranose (**g**), β-D-glucopyranose (**h**), and sucrose (**i**) into the fructose-bound structure of BmGr9 (blue, top) or unbound structure (white, bottom).

**Extended Data Table 1 | Cryo-EM data collection, refinement, and model statistics**

| | BmGr9 | BmGr9 + β-D-Fructopyranose β-D-Fructofuranose | BmGr9 + α-L-Sorbopyranose |
|---|---|---|---|
| | EMDB-42628 PDB 8UVT | EMDB-42629 PDB 8UVU | EMDB-43548 PDB 8VV3 |
| **Data collection and processing** | | | |
| Magnification | 81,000× | 81,000× | 81,000× |
| Voltage (kV) | 300 | 300 | 300 |
| Electron exposure (e⁻/Å²) | 1.67 | 1.67 | 1.67 |
| Defocus range (µm) | −0.8 to −2.5 | −0.8 to −2.5 | −0.8 to −2.5 |
| Pixel size (Å) | 1.068 | 1.068 | 1.068 |
| Symmetry imposed | C4 | C4 | C4 |
| Initial particle images (no.) | 1,179,559 | 1,157,505 | 3,604,514 |
| Final particle images (no.) | 326,748 | 307,715 | 920,021 |
| Map resolution (Å) | 2.86 | 2.99 | 2.61 |
| FSC threshold | 0.143 | 0.143 | 0.143 |
| Map resolution range (Å) | 2.3–8.0 | 2.4–9.0 | 2.3–6.0 |
| **Refinement** | | | |
| Initial model used | n/a | n/a | n/a |
| Model resolution (Å) | 4.0 | 4.0 | 4.0 |
| FSC threshold | 0.143 | 0.143 | 0.143 |
| Map sharpening $B$ factor (Å²) | −123.6 | −126.8 | −94.6 |
| Model composition | | | |
| Non-hydrogen atoms | 12104 | 11928 | 12172 |
| Protein residues | 1544 | 1516 | 1548 |
| Ligands | 0 | 4 BDF (0.75)* 4 FRU (0.25)* | 4 SOE |
| $B$ factors (Å²) | | | |
| Protein | 34.98 | 34.65 | 32.76 |
| Ligand | n/a | 33.89 | 15.43 |
| R.m.s. deviations | | | |
| Bond lengths (Å) | 0.003 | 0.003 | 0.003 |
| Bond angles (°) | 0.474 | 0.544 | 0.514 |
| Validation | | | |
| MolProbity score | 1.70 | 1.76 | 1.75 |
| Clashscore | 8.27 | 8.69 | 8.42 |
| Poor rotamers (%) | 0 | 0 | 0 |
| Ramachandran plot | | | |
| Favoured (%) | 96.32 | 95.73 | 95.80 |
| Allowed (%) | 3.68 | 4.27 | 4.20 |
| Disallowed (%) | 0 | 0 | 0 |

n/a, not applicable

* in the fructose-bound model, both β-D-fructopyranose (BDF) and β-D-fructofuranose (FRU) are in the binding pocket, at 0.75 and 0.25 occupancy, respectively.

**Extended Data Table 2 | BmGr9 and mutant activation by D-fructose**

| | R.E. | N | $F_{max}$ | Hill Slope | $EC_{50}$ (mM) |
|---|---|---|---|---|---|
| WT | 1.00 | 34 | 0.45 (0.43 to 0.48) | 1.94 (1.48 to 2.58) | 8.71 (7.40 to 10.29) |
| GCaMP | n/a | 34 | ~0.20 | ~2.37 | ~154 |
| R86A | 0.27 | 6 | | no activity | |
| R86K | 0.48 | 6 | 0.46 (0.39 to 0.56) | ~2.45 | 10.4 (6.7 to 17.2) |
| D99A | 1.26 | 6 | | no activity | |
| D99N | 1.47 | 6 | | no activity | |
| D99E | 0.52 | 6 | | no activity | |
| D165A | 0.04 | 6 | | no activity | |
| D165N | 0.19 | 6 | | no activity | |
| D165E | 0.25 | 6 | | no activity | |
| F189A | 0.62 | 6 | | no activity | |
| F189Y | 0.86 | 6 | ~0.53 | 1.05 (0.44 to 2.15) | ~100 |
| Y190A | 0.28 | 6 | | no activity | |
| Y190F | 0.08 | 6 | ~0.88 | 0.98 (0.56 to 2.07) | ~363 |
| W193F | 0.23 | 6 | | no activity | |
| T330A | 0.10 | 6 | | no activity | |
| T330V | 0.67 | 6 | ~0.13 | ~6.22 | ~98 |
| T330S | 1.72 | 6 | ~0.14 | ~3.03 | ~100 |
| P331A | 0.85 | 6 | 0.52 (0.47 to 0.60) | 1.43 (0.87 to 2.39) | 5.5 (3.7 to 8.5) |
| Q351A | 0.25 | 6 | 0.23 (0.18 to 0.65) | 1.89 (0.60 to 5.26) | ~39.1 |
| Q351N | 0.61 | 6 | ~0.17 | ~0.71 | ~8.0 |
| Q351E | 0.12 | 6 | | no activity | |
| W354F | 0.15 | 6 | | no activity | |
| H358A | 0.55 | 6 | ~0.15 | ~1.06 | ~365 |
| H358Q | 0.27 | 6 | ~0.14 | ~1.43 | ~113 |
| F333A | 0.28 | 6 | | no activity | |
| F333Y | 0.37 | 6 | | no activity | |
| F333L | 0.34 | 6 | | no activity | |
| Y332A | 0.32 | 6 | 0.17 (0.14 to 0.23) | 0.99 (0.37 to 2.17) | 8.3 (3.5 to 41.8) |
| Y332F | 0.24 | 6 | 0.36 (0.31 to 0.43) | ~2.36 | 2.6 (1.3 to 5.0) |
| Y332L | 0.21 | 6 | | no activity | |
| Y332Q | 0.06 | 6 | | no activity | |
| Q443A | 0.19 | 6 | | no activity | |
| Q443E | 0.27 | 6 | 1.14 (1.05 to 1.23) | ~2.57 | 1.32 (0.99 to 1.75) |
| Q443N | 0.40 | 6 | | no activity | |
| Q443D | 0.31 | 6 | 0.83 (0.75 to 0.96) | 1.52 (0.94 to 2.57) | 10.0 (7.4 to 14.8) |
| Q445A | 0.42 | 6 | 0.69 (0.62 to 0.75) | ~3.78 | 2.01 (1.15 to 2.71) |
| Q445E | 0.28 | 6 | 0.57 (0.47 to 0.85) | 1.56 (0.68 to 3.75) | 15.3 (9.8 to 42.2) |
| Q445N | 0.55 | 6 | 0.45 (0.38 to 0.62) | 1.49 (0.79 to 2.76) | 20.3 (13.3 to 44.4) |
| Q445K | 0.43 | 6 | | no activity | |
| Q445R | 0.22 | 6 | | no activity | |
| I440A | 0.01 | 6 | | no activity | |
| F444A | 0.38 | 6 | | no activity | |

R.E. is the relative expression level compared to wild-type (WT) BmGr9 (from Extended Data Fig. 7c) In each gel, BmGr9 was run in parallel with a set of mutants. The number of biological replicates ($N$), and fitted values for the maximum fluorescence ($F_{max}$), Hill slope, and concentration of half-maximal activation ($EC_{50}$) from the GCaMP assay are listed; 95% confidence intervals are given in parentheses. Parameters for which 95% confidence intervals could not be accurately determined are indicated by a tilde (-). n/a, not applicable. Note: the WT fit results here are based on all WT data collected; the data presented in Figs. 1b, 2f, 4f, and 5e include only WT data collected at the same time as the other mutants or sugars shown in the figures.

# Reporting Summary

## Statistics

For all statistical analyses, confirm that the following items are present in the figure legend, table legend, main text, or Methods section.

| n/a | Confirmed | |
|---|---|---|
| ☐ | ☒ | The exact sample size ($n$) for each experimental group/condition, given as a discrete number and unit of measurement |
| ☐ | ☒ | A statement on whether measurements were taken from distinct samples or whether the same sample was measured repeatedly |
| ☐ | ☒ | The statistical test(s) used AND whether they are one- or two-sided *Only common tests should be described solely by name; describe more complex techniques in the Methods section.* |
| ☒ | ☐ | A description of all covariates tested |
| ☐ | ☒ | A description of any assumptions or corrections, such as tests of normality and adjustment for multiple comparisons |
| ☐ | ☒ | A full description of the statistical parameters including central tendency (e.g. means) or other basic estimates (e.g. regression coefficient) AND variation (e.g. standard deviation) or associated estimates of uncertainty (e.g. confidence intervals) |
| ☐ | ☒ | For null hypothesis testing, the test statistic (e.g. $F$, $t$, $r$) with confidence intervals, effect sizes, degrees of freedom and $P$ value noted *Give P values as exact values whenever suitable.* |
| ☒ | ☐ | For Bayesian analysis, information on the choice of priors and Markov chain Monte Carlo settings |
| ☒ | ☐ | For hierarchical and complex designs, identification of the appropriate level for tests and full reporting of outcomes |
| ☒ | ☐ | Estimates of effect sizes (e.g. Cohen's $d$, Pearson's $r$), indicating how they were calculated |

*Our web collection on statistics for biologists contains articles on many of the points above.*

## Software and code

Policy information about availability of computer code

| Data collection | SerialEM 4.1-beta, PTI Felix 1.42b, FLIPR Tetra |
|---|---|
| Data analysis | Relion 3.1, cryoSPARC 3.3.1, CTFFIND 4.1, MotionCor2, ChimeraX 1.7, PHENIX 1.20.1-4487, HOLE, Coot 0.9.8.1, PyMOL 2.5.0, GraphPad Prism 10.0.2, JalView 2.11.3.2, ChemDraw 22.2.0, Google Colab, AlphaFold2 |

For manuscripts utilizing custom algorithms or software that are central to the research but not yet described in published literature, software must be made available to editors and reviewers. We strongly encourage code deposition in a community repository (e.g. GitHub). See the Nature Portfolio guidelines for submitting code & software for further information.

## Data

Policy information about availability of data

All manuscripts must include a data availability statement. This statement should provide the following information, where applicable:
- Accession codes, unique identifiers, or web links for publicly available datasets
- A description of any restrictions on data availability
- For clinical datasets or third party data, please ensure that the statement adheres to our policy

The final cryo-EM maps have been deposited in the Electron Microscopy Data Bank under accession numbers EMD-42629 (bound to D-fructose), EMD-42628 (unbound), and EMD-43548 (bound to L-sorbose). The final models have been deposited in the Protein Data Bank under accession numbers 8UVU (bound to β-D-fructopyranose and β-D-fructofuranose, at 75% and 25% occupancy, respectively), 8UVT (unbound), , and 8VV3 (bound to ⊟-L-sorbopyranose). Coordinates for Orco

and MhOr5, used for structural comparisons in this paper, are deposited under accession numbers 6C70 (Orco), 7LIC (MhOr5), and 7LID (MhOr5 bound to eugenol). Source activity and binding data are provided with this paper. For all other data requests, please contact J.A.B.

# Research involving human participants, their data, or biological material

Policy information about studies with human participants or human data. See also policy information about sex, gender (identity/presentation), and sexual orientation and race, ethnicity and racism.

| | |
|---|---|
| Reporting on sex and gender | Not applicable. |
| Reporting on race, ethnicity, or other socially relevant groupings | Not applicable. |
| Population characteristics | Not applicable. |
| Recruitment | Not applicable. |
| Ethics oversight | Not applicable. |

Note that full information on the approval of the study protocol must also be provided in the manuscript.

# Field-specific reporting

Please select the one below that is the best fit for your research. If you are not sure, read the appropriate sections before making your selection.

☒ Life sciences    ☐ Behavioural & social sciences    ☐ Ecological, evolutionary & environmental sciences

For a reference copy of the document with all sections, see nature.com/documents/nr-reporting-summary-flat.pdf

# Life sciences study design

All studies must disclose on these points even when the disclosure is negative.

| | |
|---|---|
| Sample size | No calculations were performed to determine sample sizes; however, the addition of more data did not alter conclusions from this study. |
| Data exclusions | No activity or binding data were excluded. Some cryo-EM images and particles were excluded from the final data set following established protocols, as described in the Methods. |
| Replication | GCaMP activity experiments were repeated on different days (> 4), using independently transfected cells and independently prepared sugar/ sweetener solutions. Trp fluorescence experiments were repeated on different days, using independently purified samples and independently prepared sugar solutions. Activity and binding data were successfully replicated each time. Structure determinations were carried out using a single data set for each sample (each independently collected over 2 days). Each cryo-EM dataser was split into two subsets and independently used to refine and validate the final model, following established protocols, as described in the Methods. |
| Randomization | This study did not allocate experimental groups; thus, no randomization was necessary. |
| Blinding | No blinding was used; all activity and binding data were analyzes using the same methods and all data were included in the results. |

# Reporting for specific materials, systems and methods

We require information from authors about some types of materials, experimental systems and methods used in many studies. Here, indicate whether each material, system or method listed is relevant to your study. If you are not sure if a list item applies to your research, read the appropriate section before selecting a response.

## Materials & experimental systems

| n/a | Involved in the study |
|---|---|
| ☐ | ☒ Antibodies |
| ☐ | ☒ Eukaryotic cell lines |
| ☒ | ☐ Palaeontology and archaeology |
| ☒ | ☐ Animals and other organisms |
| ☒ | ☐ Clinical data |
| ☒ | ☐ Dual use research of concern |
| ☒ | ☐ Plants |

## Methods

| n/a | Involved in the study |
|---|---|
| ☒ | ☐ ChIP-seq |
| ☒ | ☐ Flow cytometry |
| ☒ | ☐ MRI-based neuroimaging |

## Antibodies

| | |
|---|---|
| Antibodies used | anti-GFP AlexaFluor-488 (ThermoFisher A21311), Goat anti-rabbit IgG-HRP (ThermoFisher A11357) |
| Validation | All antibodies were from commercial sources; none were independently validated. |

## Eukaryotic cell lines

Policy information about cell lines and Sex and Gender in Research

| | |
|---|---|
| Cell line source(s) | Sf9 (ATCC CRL-1711). HEK293S GnTi– (ATCC CRL-3022), HEK293 (ATCC CRL-1573) |
| Authentication | Cell lines were obtained from commercial sources; none were independently authenticated. |
| Mycoplasma contamination | Cell lines were not tested for mycoplasma contamination. |
| Commonly misidentified lines (See ICLAC register) | Not applicable. |

## Plants

| | |
|---|---|
| Seed stocks | Not applicable. |
| Novel plant genotypes | Not applicable. |
| Authentication | Not applicable. |

