## [Peer Review File · Nature]

Manuscript Title: The molecular basis of sugar detection by an insect taste receptor

Reviewer Comments & Author Rebuttals

Reviewer Reports on the Initial Version:

Referees' comments:

Referee #1 (Remarks to the Author):

This study by Gomes et al. uses structural and functional approaches to understand how the insect fructose-gated ion channel, BmGr9, is so selective among chemically similar sugars, and how fructose triggers channel opening. They present the first cryo-EM structures of the receptor at ~3Å resolution, in fructose-bound and apo conformations. They find that fructose binds to a pocket that is conserved in location among insect ORs. The density for the agonist fructose is strong but ambiguous in the precise pose of the ligand, and also in whether it binds as the pyranose or furanose form. Docking tests are performed to rank the likelihood of different poses and for 5- vs. 6-membered ring forms of the sugar. A fluorescent calcium sensor assay is coupled with BmGr9 mutagenesis in the pocket and along the presumed gating pathway to interrogate determinants for agonist binding and channel activation. Overall the writing and figures are clear and the structural data appear solid. I found one major point to be confusing, related to the main mystery about this receptor/channel: how is it so selective for fructose? While the study presents a logical mechanism for channel gating that itself is of high interest, and appears generally conserved among insect ORs, this fundamental question about specificity is not satisfyingly addressed. I elaborate below on this one major issue, before highlighting other aspects that I think could be stronger or clearer.

1. A central theme of the study relates to what the authors refer to as a precise geometric arrangement of the binding pocket to explain why only fructose can activate the channel. However, this concept is undermined by the presented data. First, the experimental density map is ambiguous as to the orientation of fructose, which form of fructose is bound, and thus what the specific/important interactions are in the binding pocket. Second, the docking energetics for multiple poses and for furanose vs. pyranose are relatively flat. Third, docking scores for antagonists are similar to those for agonist(s). These findings suggest that precise interactions are in fact not important, and that the agonist instead can adopt multiple orientations in the pocket and that both pyranose and furanose can bind. That the channel conformation is 'open' when this fuzzy structural distribution of the ligand is bound suggests, at least superficially, that the multiple poses and ring compositions suggested by the data are efficacious in opening the channel. Do the authors agree? It is challenging to reconcile multiple agonist binding conformations and active 5- vs. 6-membered ring forms with the profound selectivity for fructose over sorbose, for example. Suggests to me that something substantial in the ligand selectivity mechanism is still not understood. I am curious what the authors think- in my examination of the data, I do not see strong support for the central theme of precise ligand coordination explaining the exquisite selectivity for agonist (fructose) vs. antagonist (sorbose). It is possible that in depth molecular dynamics simulations and/or structure(s) with antagonist(s) bound could help to connect stability of the open channel conformation to specific

ligand binding poses and specific ligand-protein interactions, and rank the strength of specific interactions.

2. Lines 82-85 imply that structures are going to be provided with both furanone and pyranose sugars bound, but only the latter is shown in Table 1. Please be explicit in your language about the composition of final model(s).

3. In looking at the ligand density in ED Fig 4 and in the experimental map (thank you for supplying), the shape could perhaps be described as heart shaped, or V-shaped, in the orientation in ED Fig 4. In the selected pose from docking, the hydroxymethyl points up and to the right in this figure. I was surprised to not see a docked pose where the hydroxymethyl is oriented into the upper left. Playing in Coot, flipping the ligand did not result in substantial changes in the binding pocket, and in both cases, it appears that some electrostatic interactions are added or subtracted but one is not obviously better than the other. Did you consider this alternate pose? For example, Gln351 does not form favorable interactions in the supplied pdb, surprising given the strong connecting density with the ligand. In the suggested alternative, that Gln side chain would form a tight interaction with the ligand hydroxymethyl. Perhaps it can bind in multiple orientations?

4. Related to uncertainty or heterogeneity in ligand-protein interactions, I am not currently convinced that it is appropriate to draw the interaction lines in Fig. 2C, D. Related, here and in Fig. 5, ligand density needs to be shown to more transparently document confidence in ligand position.

5. The unbound map supplied is very noisy in the region of residues 77-88. This is an important part of the structure, as it appears to dip into and occlude the agonist site or access to it. The atomic model may be correct, but a better map for this region would strengthen this interpretation. I presume the supplied map is globally sharpened with a single B factor. Does local sharpening or an unsharpened map reveal continuous backbone density for this important region?

6. A perhaps naïve question from someone outside of the insect OR field, why use the GCaMP assay rather than patch clamp or TEVC to more directly measure ion channel function?

7. Related to the GCaMP assays, are there controls to demonstrate that all the mutant channels make it to the cell surface? The loss of function mutants in the binding pocket are used to conclude that these residues are important for ligand binding, but this conclusion depends on whether the channel is properly folded and trafficked.

8. Section starting on line 131 emphasizes cooperativity among subunits. What is the evidence for this cooperativity? Data in Fig. 1 were used to conclude a Hill coefficient of 2-3, but this is consistent, I think, with GCaMP6 itself (PMID: 28182677), and may be unrelated to BmGr9.

9. Lines 182-191. Discussion of lipids needs to reference a figure that better shows the density and states clearly that these densities could come from detergent rather than lipids. Or, justify why the authors believe these are lipids rather than detergent. In the fructose-bound model, there are very strong lipid or detergent-like densities packed at subunit interfaces (near V428). How concerned should we be about detergent artifacts? If the other insect taste receptors have similar structures,

can a comparison of this region help us understand whether these are lipids vs. detergent, and whether they are interesting or concerning?

10. Line 236 “we have determined the first structures of a eukaryotic taste receptor” reads as if implying that the results are relevant to human sweet taste reception. Please be more precise in this wording- there are many interesting things to focus on, but it seems a stretch as currently worded. The title also has this same implication, that we are learning about pleasurable sweetness, rather than nutrient sensing. Relatedly, this whole first paragraph of the discussion relates to the pleasure of sweets- is this something for which there is evidence in invertebrates? Safer, perhaps, to focus on nutrient sensing.

11. A main finding in the later part of the study relates to many sugars being predicted to bind (docking to the new structures) while these ligands are not in fact active. Among these predicted antagonists, only sorbose and sucrose are tested in Fig. 6 and ED Fig. 9. Little insight is given into why sorbose can bind but not activate, as it appears to fit well in the bound-conformation pocket. Would additional experimental structures or more experimental binding assays lend more detailed insights into the mechanism? It is still unclear, after this figure, why the sugars predicted to bind do not activate.

Minor:

1. In an early structural figure, helices need to be labeled for a subunit; especially confusing what “S7b” means.

2. Lines 143-145, glutamate and glutamine should both enable this interaction, unclear what the basis for the enhanced activity is with the Glu substitution.

3. Fig. 4E and 5D, have any statistical tests been performed?

4. Fig 6a,b needs label the residues.

5. For interactions between amino acids or with the ligand, density needs to be shown somewhere to support confidence in these detailed interpretations.

6. ED Fig. 7A, open pore panel. Is there a mislabeling with Q445 vs Q443? In the model, Q445 orients away from the pore axis, and has no side chain density, while Q443 orients into the pore, with nice density. These are oppositely oriented in the figure panel.

7. In the unbound model, Q443 is modeled out of the density but the density looks strong. Please check its modeling.

8. Line 174 refers to a hydrogen bond that is not together with any figure. Line 184 refers to interactions not shown in Fig. 5F and G.

9. The pdb validation report highlights bond length and bond angle outliers for fructose. Please double check these.

Referee #2 (Remarks to the Author):

Elucidating the structural basis of tastants/odorants recognition by taste and olfactory receptors is fundamental and crucial for our understanding of the chemical senses. Given the diverse classes of ligands and multiple types of receptors in olfaction and taste, however, a single structure is merely a starting point in exploring this relationship; further research involving comparison of multiple receptor-ligand interactions is critically important to understand chemical detection and discrimination.

Overall, this manuscript is exciting and highly significant for the field of chemoreception, and its implications should extend further in neuroscience, nutrition, metabolism and ecology. It presents the first atomic-level description of a sugar receptor in any organism, studying BmGr9 and its comparison to known olfactory-type ion channels MhOR5 and OrCo. The authors utilize the experimentally derived structure to hypothesize about ligand-gated ion channel activation, identifying key structural elements that induce conformational changes, leading to receptor activation, and demonstrating how small structural differences in sugars significantly impact this activation. Their model, which accounts for both ligand binding and its effect on ion channel opening, introduces a novel way of thinking in this field. This manuscript advances our knowledge of chemoreception and has applications in broader fields. Its importance will undoubtedly be recognized by scientists in the field.

My suggestions for improvement are as follows:

1. Figure 1: In A, the stated differences in red are actually in blue.
2. Figure 1: Please clarify whether L-sorbose doesn't activate BmGr9 at all or acts as a partial agonist, using statistical analysis for clarity.
3. Figure 2: It would be desirable to show interactions of monosaccharides with the binding pocket in molecular dynamic simulations to better illustrate D-fructose's position, stability and interactions with specific residues.
4. Figure 3: Please label some helices in at least one domain in A and B for better orientation in parts C, D, and E.
5. Figure 4: It would be informative to investigate if L-sorbose activation changes with Q443, F444, and Q445 mutants to assess if fructose-specific activity persists.
6. Figure 6: I suggest to add a panel or two visualizing and clarifying the author's model to explain D-fructose's binding and activation of BmGr9 versus L-sorbose's binding without activation.
7. Figure 6: Please label relevant residues in parts A and B, highlighting key differences between them.
8. It would be informative to conduct a competition assay between D-fructose and L-sorbose using the GCaMP assay to test the authors' model (L-sorbose binding without activating BmGr9), which should exhibit competitive antagonism.

Author Rebuttals to Initial Comments:

Response to Referees

Gomes *et al.*, “The molecular basis of sugar detection by an insect taste receptor”

Manuscript No. 2023-11-20162

We thank the Referees for their overall enthusiasm for this work and their detailed and thoughtful comments. The manuscript has benefitted significantly from their suggestions. We are resubmitting a revised version that addresses the Referees’ concerns through additional structural and biochemical data and clarifications of the text. The key addition to the revised manuscript is a new structure of BmGr9 bound to a non-activating sugar. This structure, with a bound ligand but closed pore, allows us to uncouple structural changes associated with ligand binding from those leading to channel activation. These new data experimentally confirm our model for how sugar binding specificity is achieved at the molecular level.

Our manuscript focuses on an exemplary gustatory receptor, BmGr9 from the silk moth, which is activated only by one type of sugar, D-fructose. We determined structures of BmGr9 in multiple gating states: closed, in the absence of sugar, and opened, in the presence of D-fructose. These structures, along with mutagenesis and functional assays, illustrate how specificity for D-fructose is seemingly achieved by a ligand-binding pocket that precisely matches the overall shape and pattern of the chemical groups in the sugar. However, our computational docking and experimental binding assays revealed that other sugars also bind BmGr9, yet they are unable to activate the receptor. In our original manuscript, we proposed a mechanism for D-fructose selectivity whereby only D-fructose can both fit into the pocket and simultaneously induce a conformational change in BmGr9, thereby coupling ligand binding to pore opening.

The principal concern of the Referee’s was the lack of support for our proposed mechanism of specificity for D-fructose. In this revised version, we present the experimentally determined structure of BmGr9 bound to a non-activating sugar, L-sorbose, which has allowed us to confirm and further refine our model for selectivity. In this new structure, L-sorbose is clearly situated in the sugar-binding pocket, but it does not interact with Phe333 of the aromatic bridge. Thus, there is no link between L-sorbose in the binding pocket and the pore helix, leaving the ion conducting pore in a closed conformation. These new data confirm our original hypothesis that the precise geometric arrangement of the ligand-binding pocket is not sufficient to adequately explain why only D-fructose can activate the channel. Instead, a unique hydrophobic patch in D-fructose allows only this sugar to interact with Phe333. This hydrophobic interaction is critical for repositioning the aromatic bridge in BmGr9 (composed of Tyr332 and Phe333) closer to the bound sugar, thereby creating space for the pore helix to open.

Point-by-point responses to the Referees’ comments follow below.

Referee #1

This study by Gomes et al. uses structural and functional approaches to understand how the insect fructose-gated ion channel, BmGr9, is so selective among chemically similar sugars, and how fructose triggers channel opening. They present the first cryo-EM structures of the receptor at ~3Å resolution, in fructose-bound and apo conformations. They find that fructose binds to a pocket that is conserved in location among insect ORs. The density for the agonist fructose is strong but ambiguous in the precise pose of the ligand, and also in whether it binds as the pyranose or furanose form. Docking tests are performed to rank the likelihood of different poses and for 5- vs. 6- membered ring forms of the sugar. A fluorescent calcium sensor assay is coupled with BmGr9 mutagenesis in the pocket and along the presumed gating pathway to interrogate determinants for agonist binding and channel activation. Overall the writing and figures are clear and the structural data appear solid. I found one major point to be confusing, related to the main mystery about this receptor/channel: how is it so selective for fructose? While the study presents a logical mechanism for channel gating that itself is of high interest, and appears generally conserved among insect ORs, this fundamental question about specificity is not satisfyingly addressed. I elaborate below on this one major issue, before highlighting other aspects that I think could be stronger or clearer.

1. A central theme of the study relates to what the authors refer to as a precise geometric arrangement of the binding pocket to explain why only fructose can activate the channel. However, this concept is undermined by the presented data.

We apologize for the confusion caused by our original manuscript. We have altered our text to more clearly state our central finding that the precise geometric arrangement of the ligand-binding pocket is not sufficient to explain why only D-fructose can activate the channel. Indeed, we have identified a central switch (the aromatic bridge, composed of Tyr332 and Phe333) that couples ligand binding to pore opening. In our model for selectivity, only D-fructose can both fit into the pocket and simultaneously induce a conformational change in the aromatic bridge, thereby coupling ligand binding to pore opening. This model is further supported by our new structural data of BmGr9 bound to a non-activating sugar, L-sorbose. Sorbose is clearly situated in the sugar-binding pocket, but does not interact with Phe333 (see Fig. 6). As a consequence, the aromatic bridge does not shift and the pore remains in a closed conformation.

First, the experimental density map is ambiguous as to the orientation of fructose, which form of fructose is bound, and thus what the specific/important interactions are in the binding pocket. Second, the docking energetics for multiple poses and for furanose vs. pyranose are relatively flat. Third, docking scores for antagonists are similar to those for agonist(s). These findings suggest that precise interactions are in fact not important, and that the agonist instead can adopt multiple orientations in the pocket and that both pyranose and furanose can bind. That the channel conformation is 'open' when this fuzzy structural distribution of the ligand is bound suggests, at least superficially, that the multiple poses and ring compositions suggested by the data are efficacious in opening the channel. Do the authors agree?

We agree it is likely that both ring forms of D-fructose are able to bind and open the channel. Thus, fructose density in our bound is likely a superposition reflecting the relative natural abundance of β -D-fructopyranose (~75%) and β -D-fructofuranose (~25%). Although multiple forms of D-fructose likely activate the channel, we do not believe that there is a “fuzzy” distribution of bound ligand positions. Indeed, our experimental density is quite flat and we clearly observe asymmetric features, likely representing the positions of the hydroxymethyl arms of fructose. Please see our answer to point 3 below for further discussion about our fructose- and sorbose-bound models.

However, we would like to highlight that, despite small structural differences between the two conformers, both ring conformations share a similarly situated hydrophobic patch that interacts with Phe333. This hydrophobic interaction appears critical for repositioning the aromatic bridge (composed of Tyr332 and Phe333) closer to the bound sugar, which allows the pore to open. This model is supported by the new sorbose-bound structure: L-sorbose binds in the sugar-binding pocket but does not interact with Phe333, likely because a hydroxyl group in L-sorbose faces Phe333, disrupting a potential hydrophobic interaction, leaving the pore closed.

It is challenging to reconcile multiple agonist binding conformations and active 5- vs. 6-membered ring forms with the profound selectivity for fructose over sorbose, for example. Suggests to me that something substantial in the ligand selectivity mechanism is still not understood. I am curious what the authors think- in my examination of the data, I do not see strong support for the central theme of precise ligand coordination explaining the exquisite selectivity for agonist (fructose) vs. antagonist (sorbose).

The remarkable selectivity for D-fructose over such similar sugars as L-sorbose is what initially interested us in BmGr9. We agree that the precise ligand coordination is not sufficient to explain selectivity in BmGr9, and we have made this more explicit throughout the revised manuscript. We believe the substantial missing link to describe the remarkable selectivity for D-fructose is the aromatic bridge that connects the ligand-binding pocket to the ion channel pore. Our model for selectivity, in which only D-fructose is capable of interacting with Phe333 via a unique hydrophobic patch, makes the following predictions (1) L-sorbose, a non-activating sugar, would bind in the pocket of BmGr9, but (2) unlike D-fructose, L-sorbose would fail to interact with the aromatic bridge, leaving the pore closed. Our new structural data confirm both of these predictions, therefore validating a selectivity model whereby an activating sugar needs to bind as well as orchestrate a specific conformational change to activate the channel.

It is possible that in depth molecular dynamics simulations and/or structure(s) with antagonist(s) bound could help to connect stability of the open channel conformation to specific ligand binding poses and specific ligand-protein interactions, and rank the strength of specific interactions.

Please see our response to point 3 from Referee #2 below.

2. Lines 82-85 imply that structures are going to be provided with both furanone and pyranose sugars bound, but only the latter is shown in Table 1. Please be explicit in your language about the composition of final model(s).

We have added an additional PDB entry (8VV9) that contains the bound furanose form of D-fructose.

3. In looking at the ligand density in ED Fig 4 and in the experimental map (thank you for supplying), the shape could perhaps be described as heart shaped, or V-shaped, in the orientation in ED Fig 4. In the selected pose from docking, the hydroxymethyl points up and to the right in this figure. I was surprised to not see a docked pose where the hydroxymethyl is oriented into the upper left. Playing in Coot, flipping the ligand did not result in substantial changes in the binding pocket, and in both cases, it appears that some electrostatic interactions are added or subtracted but one is not obviously better than the other. Did you consider this alternate pose? For example, Gln351 does not form favorable interactions in the supplied pdb, surprising given the strong connecting density with the ligand. In the suggested alternative, that Gln side chain would form a tight interaction with the ligand hydroxymethyl. Perhaps it can bind in multiple orientations?

We thank the Referee for their careful examination of our model and density. We did not see a docked conformation of D-fructose with the hydroxymethyl pointed toward Gln351 (“upper left”). One reason may be the more favourable interactions that occur when the hydroxymethyl abuts Trp354. Interactions between aromatic side chains and hydrophobic faces strongly favours certain binding poses and we suggest that the interaction between D-fructose (and now, also L-sorbose) and Trp354 is an important example of such an interaction.

Furthermore, based on our new structure of BmGr9 bound to L-sorbose, we can make more definitive parallels between our structural models and docking experiments. While D-fructose exists as a five-membered and six-membered ring conformations characterized by two or one hydroxymethyl groups respectively, L-sorbose exists solely in a six-membered ring configuration with one hydroxymethyl group. In our new structure, the density for L-sorbose has a single “arm” facing to the “right” (Fig. 6); the resulting model closely resembles the docked position of L-sorbose and suggests that our original model of the six-membered ring of D-fructose in a similar pose is correct. In light of this new structural data with a single arm of sugar density, the “heart shaped” density of D-fructose likely is the result from a superposition of both the five- and six-membered ring forms. The five-membered β -furanose ring form has two hydroxymethyl arms that fit nicely in the density, with density surrounding the single arm to the right being more predominant because of the contribution from both ring types. We have added text to clarify this point (see Extended Data Fig. 4) and included an additional PDB entry with the refined fructofuranose structure.

4. Related to uncertainty or heterogeneity in ligand-protein interactions, I am not currently convinced that it is appropriate to draw the interaction lines in Fig. 2C, D. Related, here and in Fig. 5, ligand density needs to be shown to more transparently document confidence in ligand position.

We have added ligand density to Figs. 2 and 5. In both cases (fructose and sorbose), asymmetric features are present in the density, which allows us to confidently model the bound sugars. Furthermore, we have also created a new Extended Data Figure (ED Fig. 5) that shows density for the ligands and binding residues. Please, see our response to point 3 for further discussion about our confidence in modelling the ligand positions and interacting residues.

5. The unbound map supplied is very noisy in the region of residues 77-88. This is an important part of the structure, as it appears to dip into and occlude the agonist site or access to it. The atomic model may be correct, but a better map for this region would strengthen this interpretation. I presume the supplied map is globally sharpened with a single B factor. Does local sharpening or an unsharpened map reveal continuous backbone density for this important region?

We have updated our map and model for unbound BmGr9. This map clearly shows continuous density in this region (albeit at a slightly lower contour level). Likely, this loop is less ordered in the apo state and Arg86 anchors it in the bound structure, resulting in improved density for this region in our bound structures.

6. A perhaps naïve question from someone outside of the insect OR field, why use the GCaMP assay rather than patch clamp or TEVC to more directly measure ion channel function?

In our study, we extensively use mutagenesis to interrogate the roles of specific amino acids in BmGr9 function. The GCaMP assay has been widely used in this field (see Butterwick *et al. Nature*, 2018 and

del Marmol *et al. Nature*, 2021) as it allows us to assess channel activity in a high-throughput manner. Although finer details of ion channel kinetics may not be apparent, this assay robustly measures channel activity, which is our main purpose for the present study.

We do agree, however, that electrophysiological measurements of ion channel activity would be an excellent complementary technique to investigate detailed effects of these mutations for a follow-up study.

7. Related to the GCaMP assays, are there controls to demonstrate that all the mutant channels make it to the cell surface? The loss of function mutants in the binding pocket are used to conclude that these residues are important for ligand binding, but this conclusion depends on whether the channel is properly folded and trafficked.

The Referee makes an excellent point. We have added new data where we extracted wild-type and mutant BmGr9 receptors and used Native PAGE to assess tetramer formation. We have included gel images in Extended Data Figs. 6 and 8, and quantified expression levels in Extended Data Table 2. Importantly, there is no clear relationship between expression level and activity in our assay.

8. Section starting on line 131 emphasizes cooperativity among subunits. What is the evidence for this cooperativity? Data in Fig. 1 were used to conclude a Hill coefficient of 2-3, but this is consistent, I think, with GCaMP6 itself (PMID: 28182677), and may be unrelated to BmGr9.

We note that our observation of a large Hill coefficient (2.2) agrees with previous work on BmGr9 in oocytes, where electrophysiological experiments measured a Hill coefficient of 1.7 (Morinaga *et al. J. Biol. Chem.*, 2022).

But the Referee raises another excellent point about the role of the intrinsic Hill coefficient of GCaMP itself. In previous work on insect olfactory receptors, a wide range of Hill coefficients have been measured using the GCaMP assay, from 1 to ~3 (see Butterwick *et al. Nature*, 2018, and del Marmol *et al. Nature*, 2021). In our manuscript, we observed a similar range of Hill coefficients, with many mutant receptors having a coefficient near 1 (see Extended Data Table 2). These observations suggest that the intrinsic Hill coefficient for GCaMP6s (~3, measured by Helassa *et al. Sci. Rep.*, 2016) does not limit what is detected by the GCaMP assay. We suspect that the influx of Ca^{2+} from outside of the cell (~5 mM) locally saturates GCaMP6s ($K_d = 0.1 \mu\text{M}$), making the change in fluorescence proportional to the number of open channels.

9. Lines 182-191. Discussion of lipids needs to reference a figure that better shows the density and states clearly that these densities could come from detergent rather than lipids. Or, justify why the authors believe these are lipids rather than detergent. In the fructose-bound model, there are very strong lipid or detergent-like densities packed at subunit interfaces (near V428). How concerned should we be about detergent artifacts? If the other insect taste receptors have similar structures, can a comparison of this region help us understand whether these are lipids vs. detergent, and whether they are interesting or concerning?

We have modified our discussion about the potential source of these columnar densities to include detergent. We have also moved the figures describing these densities to Extended Data Fig. 10, which now have additional panels more clearly describing the changing interaction surrounding this region between the closed (unbound) and open (fructose-bound) states.

Thus far, the only published structures of insect olfactory receptors were determined in detergent.

[This has been redacted.]

10. Line 236 “we have determined the first structures of a eukaryotic taste receptor” reads as if implying that the results are relevant to human sweet taste reception. Please be more precise in this wording- there are many interesting things to focus on, but it seems a stretch as currently worded. The title also has this same implication, that we are learning about pleasurable sweetness, rather than nutrient sensing. Relatedly, this whole first paragraph of the discussion relates to the pleasure of sweets- is this something for which there is evidence in invertebrates? Safer, perhaps, to focus on nutrient sensing.

Our title explicitly mentions that our work focuses on an “insect” taste receptor, which should mitigate any confusion with work on human sweet taste perception. Insects do, however, exhibit hallmarks of a pleasurable sensation when eating sugar. In both mammals and insects, pairing a neutral stimulus with sugar, induces the emergence of an appetitive response towards the previously neutral stimulus. Sugar acts by directly activating reward circuits, such as the dopaminergic neurons that target the learning center in *Drosophila*, a neural mechanism that is conserved in mammals as well.

11. A main finding in the later part of the study relates to many sugars being predicted to bind (docking to the new structures) while these ligands are not in fact active. Among these predicted antagonists, only sorbose and sucrose are tested in Fig. 6 and ED Fig. 9. Little insight is given into why sorbose can bind but not activate, as it appears to fit well in the bound-conformation pocket. Would additional experimental structures or more experimental binding assays lend more detailed insights into the mechanism? It is still unclear, after this figure, why the sugars predicted to bind do not activate.

We have included additional structural data that shows L-sorbose bound in the binding pocket, but unable to activate the channel. See our answer to point 1 above, that describes why L-sorbose cannot activate BmGr9 when bound.

Minor:

12. In an early structural figure, helices need to be labeled for a subunit; especially confusing what “S7b” means.

Helices have been labelled beginning in Fig. 2. In the first structure of a member of this superfamily (Orco), the last transmembrane helix was found to contain two helices (called S7a and S7b) connected by a beta-hairpin loop. S7a is in the anchor domain and S7b is the pore helix. Since this nomenclature is not widely known, we have also included descriptors throughout the text to help the reader (i.e., “pore helix” for S7b).

13. Lines 143-145, glutamate and glutamine should both enable this interaction, unclear what the basis for the enhanced activity is with the Glu substitution.

The mechanism underlying the enhancement of activity when Gln443 is mutated to Glu is not known, and may not only be due to change in the interaction with Tyr332. Indeed, several other publications have noted similar increases in activity when pore residues are mutated in insect olfactory receptors (see Butterwick *et al. Nature*, 2018, and del Marmol *et al. Nature*, 2021), suggesting that the amino acid

identities of pore-lining residues strongly affect the balance between open and closed states of the channel.

14. Fig. 4E and 5D, have any statistical tests been performed?

Statistical tests have been performed on these data and are now reported in the corresponding figures and legends.

15. Fig 6a,b needs label the residues.

Docking models originally presented in Fig. 6a,b have been replaced by data from an experimentally determined structure of BmGr9 bound to L-sorbose. Interacting residues have been labelled.

16. For interactions between amino acids or with the ligand, density needs to be shown somewhere to support confidence in these detailed interpretations.

We have added a new Extended Data Figure (ED Fig. 5), which highlights density of the ligand and interacting residues.

17. ED Fig. 7A, open pore panel. Is there a mislabeling with Q445 vs Q443? In the model, Q445 orients away from the pore axis, and has no side chain density, while Q443 orients into the pore, with nice density. These are oppositely oriented in the figure panel.

We thank the reviewer for this careful observation. Yes, the original figure was mislabeled and it has been corrected.

18. In the unbound model, Q443 is modeled out of the density but the density looks strong. Please check its modeling.

We have further refined our unbound model and Gln443 is now within the density.

19. Line 174 refers to a hydrogen bond that is not together with any figure. Line 184 refers to interactions not shown in Fig. 5F and G.

Images showing these interactions have been added to Extended Data Fig. 5.

20. The pdb validation report highlights bond length and bond angle outliers for fructose. Please double check these.

We have further refined our fructose-bound model and have updated the coordinates.

Referee #2:

Elucidating the structural basis of tastants/odorants recognition by taste and olfactory receptors is fundamental and crucial for our understanding of the chemical senses. Given the diverse classes of ligands and multiple types of receptors in olfaction and taste, however, a single structure is merely a starting point in exploring this relationship; further research involving comparison of multiple receptor-ligand interactions is critically important to understand chemical detection and discrimination.

Overall, this manuscript is exciting and highly significant for the field of chemoreception, and its implications should extend further in neuroscience, nutrition, metabolism and ecology. It presents the first atomic-level description of a sugar receptor in any organism, studying BmGR9 and its comparison to known olfactory-type ion channels MhOR5 and OrCo. The authors utilize the experimentally derived structure to hypothesize about ligand-gated ion channel activation, identifying key structural elements that induce conformational changes, leading to receptor activation, and demonstrating how small structural differences in sugars significantly impact this activation. Their model, which accounts for both ligand binding and its effect on ion channel opening, introduces a novel way of thinking in this field. This manuscript advances our knowledge of chemoreception and has applications in broader fields. Its importance will undoubtedly be recognized by scientists in the field.

My suggestions for improvement are as follows:

1. Figure 1: In A, the stated differences in red are actually in blue.

We thank the Referee for catching this error; it has been corrected.

2. Figure 1: Please clarify whether L-sorbose doesn't activate BmGr9 at all or acts as a partial agonist, using statistical analysis for clarity.

The activation of BmGr9 by L-sorbose is not statistically different than GCaMP alone, suggesting that it does not activate BmGr9 (see Fig. 1). Furthermore, L-sorbose was also shown not to activate BmGr9 in oocytes, measured using electrophysiology (see Sato *et al. PNAS*, 2011).

3. Figure 2: It would be desirable to show interactions of monosaccharides with the binding pocket in molecular dynamic simulations to better illustrate D-fructose's position, stability and interactions with specific residues.

Our new structure of L-sorbose bound to BmGr9 aids in addressing some of these outstanding concerns: a single hydroxyl group disrupts the interactions between the sugar and the hydrophobic bridge that links the ligand-binding pocket to the pore. These data support the ligand binding pose and ligand-protein interaction motifs we have described as key for the channel to be functional.

We agree with the Referee that in-depth molecular dynamics simulations may help refine our model even further, but we believe these studies are beyond the scope of this work. Instead, we provide a detailed structural comparison between an activating and non-activating sugar, elucidating their mode of binding and how that translates (or not) into channel activity.

4. Figure 3: Please label some helices in at least one domain in A and B for better orientation in parts C, D, and E.

We have added helix labels in Fig. 3a,b, as suggested.

5. Figure 4: It would be informative to investigate if L-sorbose activation changes with Q443, F444, and Q445 mutants to assess if fructose-specific activity persists.

We agree that it would be very interesting and informative if a BmGr9 mutation could change sugar specificity. Thus far, it does not appear that L-sorbose can activate any of these BmGr9 mutants. In our structure of BmGr9 bound to L-sorbose, this sugar does not move the aromatic bridge, which may prevent any activation of BmGr9, even with mutations that seem to have increased activity. We are currently screening other non-activating sugars (and additional mutations) to identify any that might be able to activate a mutant BmGr9. The results of these studies, however, are beyond the scope of the current work.

6. Figure 6: I suggest to add a panel or two visualizing and clarifying the author's model to explain D-fructose's binding and activation of BmGr9 versus L-sorbose's binding without activation.

In Fig. 6, we have added panels showing the experimentally derived structure of BmGr9 bound to L-sorbose. This structure highlights how L-sorbose sits in the sugar-binding pocket, but does not contact Phe333 in the aromatic bridge, resulting in the pore remaining closed. Please see our response to Referee #1, point

7. Figure 6: Please label relevant residues in parts A and B, highlighting key differences between them.

Docking models originally presented in Fig. 6a,b have been replaced by data from an experimentally determined structure of BmGr9 bound to L-sorbose. Interacting residues have been labelled.

8. It would be informative to conduct a competition assay between D-fructose and L-sorbose using the GCaMP assay to test the authors' model (L-sorbose binding without activating BmGr9), which should exhibit competitive antagonism.

We now provide structural data to demonstrate that L-sorbose indeed binds BmGr9, similarly to D-fructose, but does not open the pore. These data not only support the Referee's hypothesis that there should be competitive antagonism between the two sugars, but provides mechanistic evidence that the channel selectivity requires sugar binding as well as the engagement of the aromatic bridge to open the pore.

Reviewer Reports on the First Revision:

Referees' comments:

Referee #1 (Remarks to the Author):

I am satisfied with the responses from the authors to my comments on the first version, including text revisions and the new data, which are exciting. I also want to clarify (correct) my comment about the title- I was in error, your title is fine.

Referee #2 (Remarks to the Author):

The revised manuscript has surpassed my initial expectations. The inclusion of the structure of BmGr9 bound to L-sorbose compellingly supports the manuscript's central point, shedding light on the mechanism by which BmGr9 discriminates between sugar epimers. This structure not only elucidates the critical components necessary for the specific sugar-receptor binding but also demonstrates how the interaction with a specific type of bound sugar triggers structural alterations, facilitating the opening of the receptor pore. The manuscript offers vital insights into the mechanism of ligand-gated activation, relevant to insect gustatory receptors and beyond. I am satisfied with the modifications made in this revised version.

Additionally, I wish to discuss the recent publication by D. Ma et al. in *Science* (2024), titled "Structural basis for sugar perception by *Drosophila* gustatory receptors," which also explores the binding of sugars to insect GRs. This study, focusing on two *Drosophila* GRs, employs experimentally determined structures to investigate sugar binding and activation mechanisms across various mono- and disaccharides. While this paper is of high quality, it does not intrude the novel contributions of the BmGr9 study, particularly regarding the fine-tuning mechanism of BmGr9. The BmGr9 manuscript uniquely advances our understanding by describing a specific mechanism of receptor tuning, distinguishing it from the focus on more coarse mechanisms of sugar binding observed in the work by D. Ma et al.